



Hydrology and Earth System Sciences

# Technical note: Multi-objective calibration by combination of stochastic and gradient-like parameter generation rules – the caRamel algorithm

**Céline Monteil**[1], **Fabrice Zaoui**[1], **Nicolas Le Moine**[2], and **Frédéric Hendrickx**[1]

[1]EDF R&D LNHE – Laboratoire National d'Hydraulique et Environnement, Chatou, 78400, France
[2]UMR 7619 Metis (SU/CNRS/EPHE), Sorbonne Université, 4 Place Jussieu, Paris, 75005, France

**Correspondence:** Céline Monteil (celine-c.monteil@edf.fr)

**Abstract.** TS1 Environmental modelling is complex, and models often require the calibration of several parameters that are not able to be directly evaluated from a physical quantity or field measurement. Multi-objective calibration has many advantages such as adding constraints in a poorly constrained problem or finding a compromise between different objectives by defining a set of optimal parameters. The caRamel optimizer has been developed to meet the requirement for an automatic calibration procedure that delivers not just one but a family of parameter sets that are optimal with regard to a multi-objective target. The idea behind caRamel is to rely on stochastic rules while also allowing more "local" mechanisms, such as the extrapolation along vectors in the parameter space. The caRamel algorithm is a hybrid of the multi-objective evolutionary annealing simplex (MEAS) method and the non-dominated sorting genetic algorithm II ($\varepsilon$-NSGA-II). It was initially developed for calibrating hydrological models but can be used for any environmental model. The caRamel algorithm is well adapted to complex modelling. The comparison with other optimizers in hydrological case studies (i.e. NSGA-II and MEAS) confirms the quality of the algorithm. An R package, caRamel, has been designed to easily implement this multi-objective algorithm optimizer in the R environment.

## 1 Introduction

Environmental modelling is complex, and models often require the calibration of many parameters that cannot be directly estimated from a physical quantity or a field measurement. Moreover, as models' outputs exhibit errors whose statistical structure may be difficult to characterize precisely, it is frequently necessary to use various objectives to evaluate the modelling performance. In other words, it is often difficult to find a rigorous likelihood function or sufficient statistics to be maximized/minimized (Fisher, 1922); for example, it is well known that errors in a simulated discharge time series are not normally distributed, and do not have constant variance or autocorrelation (Sorooshian and Dracup, 1980). In addition, Efstratiadis and Koutsoyiannis (2010) list other advantages of multi-objective calibration such as ensuring parsimony between the number of objectives and the parameters to optimize, fitting distributed responses of models on multiple measurements, recognizing the uncertainties and structural errors related to the model configuration and the parameter estimation procedure, and handling objectives that have contradictory performance.

Multi-objective calibration allows for a compromise between these different objectives to be found by defining a set of optimal parameters. Practical experience shows that single-objective calibrations are efficient for highlighting a certain property of a system, but this might lead to increasing errors in some other characteristics (Mostafaie et al., 2018). Evolutionary algorithms have been widely used to explore the Pareto-optimal front in multi-objective optimization problems that are too complex to be solved by descent

**Published by Copernicus Publications on behalf of the European Geosciences Union.**

methods with classical aggregation approaches. Evolutionary algorithms are advantageous not only because there are few alternatives for searching substantially large spaces for multiple Pareto-optimal solutions but also due to their inherent parallelism and capability to exploit similarities of solutions by recombination that enables them to approximate the Pareto-optimal front in a single optimization run (Zitzler et al., 2000).

Many studies have used the multi-objective approach in environmental modelling (Oraei Zare et al., 2012; Ercan and Goodall, 2016) or in land use models (Gong et al., 2015; Newland et al., 2018). In hydrology, Madsen (2003) implemented automatic multi-objective calibration of the MIKE SHE model (Refsgaard and Storm , 1995) on the Danish Karup catchment ($440 \, \mathrm{km}^2$) using the shuffled complex evolution (SCE) algorithm (Duan et al., 1992). Yang et al. (2014) ran a multi-objective optimization of the MOBIDIC (Campo et al., 2006) distributed hydrologic model on the Davidson catchment (North Carolina, $105 \, \mathrm{km}^2$) using the non-dominated sorting genetic algorithm II (NSGA-II, Deb et al., 2002). More recently Smith et al. (2019) led a multi-objective ensemble approach to hydrological modelling in the UK over 303 catchments for historic drought reconstruction with the GR4J conceptual model (Coron et al., 2017) using Latin hypercube sampling (McKay et al., 1979) and a Pareto-optimizing ranking approach accounting for unacceptable trade-offs (Efstratiadis and Koutsoyiannis, 2010). Mostafaie et al. (2018) compared five different calibration techniques on the GR4J lumped hydrological model using in situ runoff and daily data from the Gravity Recovery And Climate Experiment (GRACE, Tapley et al., 2004). They came to the following conclusions: according to the diversity-based metrics, the NSGA-II method is the best approach; according to the accuracy metric, multi-objective particle swarm optimization (MPSO, Reddy and Nagesh Kumar , 2007) is ranked first; and, considering the cardinality measure, the performance of all algorithms is found to be the same.

The caRamel optimizer has been developed to meet the need for an automatic calibration procedure that delivers not only one but a family of parameter sets that are optimal with regard to a multi-objective target (Le Moine, 2009). Madsen (2003) indicated that global population-evolution-based algorithms are more effective than multi-start local search procedures, which, in turn, perform better than purely local search methods. However, most of the multi-objective algorithms rely mainly on stochastic generation rules, with few deterministic aspects, as is the case in the widely used NSGA-II for instance. The idea behind caRamel is not just to keep these stochastic "global" mechanisms (such as recombination or multivariate sampling using the covariance) but also to allow more "local" mechanisms, such as extrapolation along vectors in the parameter space, that are associated with an improvement in all objective functions (a "gradient-like"

qualitative approach extended to the set of objective functions).

The caRamel algorithm was initially developed and used for the calibration of hydrological models by studies such as Rothfuss et al. (2012), Magand et al. (2014), Le Moine et al. (2015), Monteil et al. (2015), which were all previous to the R package release, or Rouhier et al. (2017), which utilized an R version for the calibration of a hydrologic model over the Loire Basin ($35\,707 \, \mathrm{km}^2$). The interesting performance of the caRamel algorithm in such studies prompted us to describe the algorithm in detail in the present paper. Considering the increasing use of R in hydrology (Slater et al., 2019), we decided to build an R package, `caRamel`, for use in any model in the R environment. The user simply has to define a vector-valued function (at least two objectives) for the model to calibrate as well as upper and lower bounds for the calibrated parameters.

This paper aims to describe the principles of the caRamel algorithm via an analysis of its results when used for the parametrization of hydrological models. Pieces of code are provided in the Appendix. For an analytical example and for three river case studies, a comparison with the two calibration algorithms that inspired caRamel, the non-dominated sorting genetic algorithm II (NSGA-II; Reed and Devireddy, 2004) and the multi-objective evolutionary annealing simplex method (MEAS; Efstratiadis and Koutsoyiannis, 2008), is also presented.

## 2 Context and notations

The intent of multi-objective calibration is to find sets of parameters that provide a compromise between several potentially conflicting objectives; for instance, how to achieve a good simulation of both flood and low-flow conditions in a hydrological model. Multi-objective calibration is also a means of adding some constraints to an under-constrained problem when many parameters have to be quantified. This can help to reduce the equifinality of parameter sets. Her and Seong (2018) showed that the introduction of an adequate number of objective functions could improve the quality of a calibration without requiring additional observations. The amount of equifinality and the overall output uncertainty decreased while the model performance was maintained as the number of objective functions increased sequentially until four objective functions.

To introduce our notation, Fig. 1 shows a simplified calibration problem in which there is

- a model with $n_\theta = 2$ parameters to calibrate ($\theta_1$ and $\theta_2$). Thus, the model structure is unequivocally represented by the vector $\boldsymbol{\theta} = (\theta_1, \theta_2)$ in a $n_\theta = 2$-dimensional space, called "parameter space $\mathcal{E}_\theta$".

- a vector $\boldsymbol{y}$ of $n_y$ observed values that should be simulated by the model. For example, for daily times series

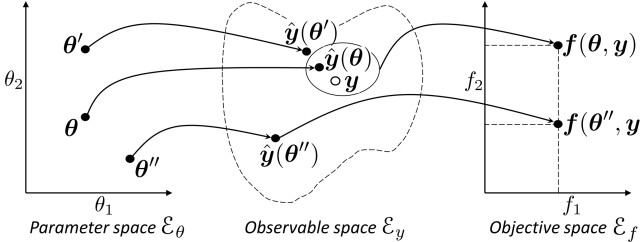

**Figure 1.** Notations to describe a model calibration, where $\boldsymbol{\theta}$ is a vector from the parameter space $\mathcal{E}_\theta$, $\boldsymbol{y}$ is a vector of observed values in the observable space $\mathcal{E}_y$ and $\boldsymbol{f}(\boldsymbol{\theta}, \boldsymbol{y})$ is an objective vector in the objective space $\mathcal{E}_f$.

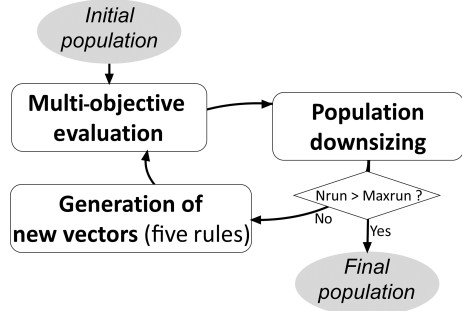

**Figure 2.** Flowchart of the caRamel algorithm.

of 1 year at two gauging stations, $n_y = 2 \times 365 = 730$. The simulation is represented by a vector $\hat{\boldsymbol{y}}(\boldsymbol{\theta})$ in a $n_y$-dimensional space (that cannot be illustrated graphically), called "observable space $\mathcal{E}_y$".

– a vector of $n_f$ objective values $\boldsymbol{f}(\boldsymbol{\theta}, \boldsymbol{y})$. For the example in Fig. 1, $\boldsymbol{f} = (f_1, f_2)$ in a space with $n_f$ dimensions, called "objective space $\mathcal{E}_f$".

We will use the following notations: vectors or matrices are presented using bold italic and bold roman font respectively ($\boldsymbol{\theta}$, $\boldsymbol{y}$, $\boldsymbol{f}$, $\boldsymbol{\Sigma}$...), vector elements and scalars are presented using roman font ($\theta_1$, $\theta_2$, $\lambda$, ...), and spaces or ensembles are presented using italic font ($\mathcal{E}_\theta$, $\mathcal{F}$, $\mathcal{A}$, ...).

Figure 1 also illustrates the relevance of multi-objective calibration with regard to two kinds of equifinality:

1. equifinality of a structure – the two points $\boldsymbol{\theta}$ and $\boldsymbol{\theta}'$ that are quite distant in the parameter space $\mathcal{E}_\theta$ become quite near in the observable space $\mathcal{E}_y$.

2. equifinality related to the objective – the vectors $\boldsymbol{\theta}$ and $\boldsymbol{\theta}''$ are equifinal regarding $f_1$, and the additional objective $f_2$ helps to discriminate them. The use of additional objectives may then help to better constrain the calibration.

The purpose of a multi-objective algorithm is to approach the Pareto front, $\mathcal{F}$, of non-dominated solution in the objective space using an ensemble of points called the approximated Pareto front $\hat{\mathcal{F}}$. We call "archive $\hat{\mathcal{A}}$" the ensemble of parameter sets from $\mathcal{E}_\theta$ for which simulation outputs are in $\hat{\mathcal{F}}$.

## 3 The caRamel algorithm description

The caRamel algorithm belongs to the genetic algorithm family. The idea is to start from an ensemble of parameter sets (called a "population") and to make this population evolve following certain generation rules (Fig. 2). At each generation, new sets are evaluated regarding the objectives, and only the more "suitable" sets are kept to build the new population. The caRamel algorithm is largely inspired by

1. the multi-objective evolutionary annealing simplex method (MEAS; Efstratiadis and Koutsoyiannis, 2005; Efstratiadis and Koutsoyiannis, 2008), with respect to the directional search method, based on the simplexes of the objective space, and

2. the non-dominated sorting genetic algorithm II ($\varepsilon$-NSGA-II; Reed and Devireddy, 2004), for the classification of parameter vectors and the management of precision by $\varepsilon$ dominance.

This section describes the functioning of the caRamel algorithm; this algorithm has been implemented in an R package, `caRamel`, that is described in Appendix A.

### 3.1 Generation rules

The caRamel algorithm has five rules for producing new solutions at each generation: (1) interpolation, (2) extrapolation, (3) independent sampling with a priori parameter variance, (4) sampling with respect to a correlation structure and (5) recombination.

The first two rules (interpolation and extrapolation) are based on a $n_\theta$-dimensional Delaunay triangulation in the objective space $\mathcal{E}_f$. They assume that two neighbouring points in the objective space $\mathcal{E}_f$ have two adjacent points in the parameter space $\mathcal{E}_\theta$ as antecedents; therefore, one can try to "guess" the directions of improvement in the parameter space from the improvement directions (in a Pareto sense) in the objective space, at least near the optimal zone.

The following two rules create new parameter sets by exploring the parameter space in a nondirectional and less local way – either by independent variations in each parameter or by multivariate sampling using the covariance structure of all parameter sets located near the estimated Pareto front at the current iteration.

Finally, the recombination rule consists of creating new parameter sets using two partial subsets derived from a pair of previously evaluated parameter sets (inspired by Baluja and Caruana, 1995).

### 3.1.1 Rule 1: interpolation

For rules 1 and 2, we use the notion of simplex which is a generalization of the notion of a triangle to higher dimensions: a 0-simplex is a point, a 1-simplex is a line segment, a 2-simplex is a triangle and a 3-simplex is a tetrahedron. A vertex is a point where two or more edges meet. The explanation of the first rule is based on Fig. 3a. First, a triangulation of the points in the objective space $\mathcal{E}_f$ is established: simplexes built with these points $\boldsymbol{f}(\boldsymbol{\theta}_i)$ are a partition of the explored zone in this space (Efstratiadis and Koutsoyiannis, 2005).

Let us consider a simplex with at least one vertex on the approximated Pareto front. This simplex is the result of the function $\boldsymbol{f}$ from an ensemble of $(n_f + 1)$ points from the $n_\theta$-dimensional parameter space $\mathcal{E}_\theta$. Under the hypothesis of continuity of $\boldsymbol{f}$, a linear combination of the form $\boldsymbol{\theta} = w_1\boldsymbol{\theta}_1 + \ldots + w_{(n_f+1)}\boldsymbol{\theta}_{(n_f+1)}$, with the barycentric coordinates $w_i \geq 0$ and $\sum_i w_i = 1$, might give a new Pareto-optimal solution $\boldsymbol{f}(\tilde{\boldsymbol{\theta}})$ inside this zone.

First the triangulation is established, then simplex volumes are computed. The probability of generating one new point with a simplex is proportional to its volume when it has at least one point on the Pareto front (otherwise it is zero). If the simplex is selected, then a set of barycentric coordinates are computed by randomly generating $(n_f + 1)$ values $\varepsilon_i$ in a uniform distribution on [0,1]:

$$w_i = \frac{\varepsilon_i}{\sum_{j=1}^{(n_f+1)} \varepsilon_j} \qquad (1)$$

### 3.1.2 Rule 2: extrapolation

Extrapolation is based on the same hypothesis of continuity as interpolation. In this case, it is tested to find if an improvement may be obtained by extrapolating from certain directions. These directions are computed from the triangulation by selecting the edges that have only one vertex on to the approximated Pareto front (the second vertex is dominated by the first). These oriented edges computed from the objective space represent directions of improvement in the parameter space (Fig. 3b).

The length $L = \|\boldsymbol{f}(\boldsymbol{\theta}_1) - \boldsymbol{f}(\boldsymbol{\theta}_2)\|$ of each selected edge and the mean length $\overline{L}$ are computed. The probability of using an edge is proportional to its length $L$. In this case, the research vector in the parameter space is defined in Eq. (2), and a new parameter set is generated by $\tilde{\boldsymbol{\theta}} = \boldsymbol{\theta}_1 + \lambda U$, where $\lambda$ is a scalar from an exponential distribution with average of 1.

$$U = \frac{L}{\overline{L}}(\boldsymbol{\theta}_1 - \boldsymbol{\theta}_2) \qquad (2)$$

### 3.1.3 Rule 3: independent sampling with a priori parameter variance

The drawback of the first two rules is that the generation of new vectors is only based on a small number of existing vec-

tors. To compensate for this search by gradient and to avoid convergence toward a local optimum, the third generation rule has two goals:

- to make the parameters vary within a larger range than with local rules, and

- to make the parameters vary independently of one another.

When considering a vector $\boldsymbol{\theta}$ from the archive $\hat{\mathcal{A}}$, the third rule is to generate $n_\theta$ new vectors ($\tilde{\boldsymbol{\theta}}_k$ with $k$ from 1 to $n_\theta$) by making each element of $\boldsymbol{\theta}$ (Eq. (3)) vary individually, where $\sigma_i^2$ is the a priori variance of the $i$th parameter and $\varepsilon_i$ is a value from a normal distribution (with an average of 0 and a variance of 1). The a priori variance is computed for each parameter from the bounds of variation indicated as the input of the optimizer.

$$\forall i \in [1:n_\theta]_{i\neq k} \quad \tilde{\theta}_{ki} = \theta_i; \quad \text{if } i = k \quad \tilde{\theta}_{ki} = \theta_{ki} + \sigma_i\varepsilon_i \quad (3)$$

The algorithm selects the $n_\theta$ vectors that maximize each element of the objective vector individually as well as an additional vector that represents a "central" point of the Pareto front. To select this vector, the minimum of each vector $\boldsymbol{\theta} \in \hat{\mathcal{A}}$ is computed, and the vector that maximizes this value is chosen.

One generation of this rule then produces $(n_f+1)\times n_\theta$ new vectors. For this reason, this rule is applied every $K$ generation, with $K$ to be defined by the user. By default, $K$ is computed so that each rule generates the same number of vectors on average.

### 3.1.4 Rule 4: sampling with respect to a correlation structure

The variance–covariance matrix $\boldsymbol{\Sigma}$ is computed using Eq. (4), where $\mathbb{E}[X]$ is the expectancy of a random variable $X$, $\boldsymbol{\theta}$ is a vector from the archive $\mathcal{A}$, $\boldsymbol{\mu} = \mathbb{E}_{\boldsymbol{\theta}\in\mathcal{A}}[\boldsymbol{\theta}]$ is the barycentre of $\mathcal{A}$ and $\mathbf{M}^T$ is the transpose of the matrix $\mathbf{M}$.

$$\boldsymbol{\Sigma} = \mathbb{E}_{\boldsymbol{\theta}\in\mathcal{A}}\left[(\boldsymbol{\theta} - \boldsymbol{\mu})(\boldsymbol{\theta} - \boldsymbol{\mu})^T\right] \qquad (4)$$

This matrix reflects the correlation structure between the parameter sets. For instance, in the case of a hydrological model, parameters are frequently not independent of each other. This rule intends to obtain an estimate $\hat{\boldsymbol{\Sigma}}$ of $\boldsymbol{\Sigma}$ and $\hat{\boldsymbol{\mu}}$ of $\boldsymbol{\mu}$ in order to generate new parameter vectors that respect this correlation structure and, therefore, limit the risk of generating "non-functional" parameter sets.

There are many possibilities in selecting the vector for evaluating the covariance matrix:

1. Vectors may be selected from a library of "historical" vectors for the calibrated model. The drawback is that this library has to be previously established, and it does not take the progression of the running calibration into account.

Hydrol. Earth Syst. Sci., 24, 1–21, 2020 https://doi.org/10.5194/hess-24-1-2020

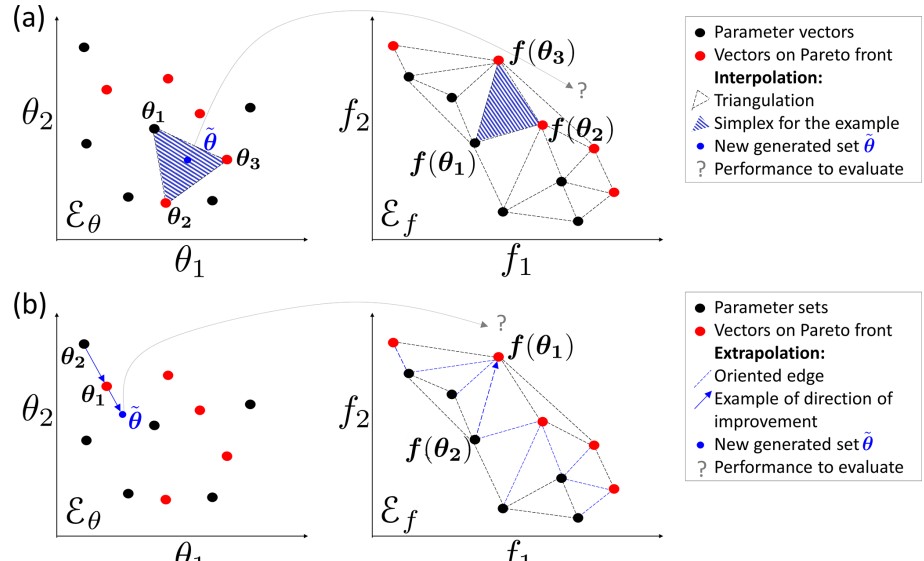

**Figure 3.** Illustration of rules 1 and 2 based on a Delaunay triangulation in the objective space for a maximization problem with two parameters ($\theta_1$ and $\theta_2$) and two objectives ($f_1$ and $f_2$): **(a)** interpolation computes a new parameter vector for each simplex with a non-dominated vertex; **(b)** extrapolation derives a new vector for each direction of improvement.

2. Vectors may be selected from the archive $\hat{\mathcal{A}}$ that provides points on the approximated Pareto front at the running generation. The new vectors frequently improve the front, but, as the variance is low, they do not avoid convergence toward a local optimum.

3. All vectors of the running population may be selected. This helps to maintain diversity, but it has a high computational cost as few new vectors will make the front progress.

Finally, the algorithm uses a mix between items 2 and 3: all simplexes from the first rule triangulation that have at least one vertex in the approximated Pareto front are selected. Reference vectors for the computation of the variance–covariance matrix are defined by the ensemble $\mathcal{G}$ from the objective space whose images by $f$ are all the vertices of these simplexes. The estimates $\hat{\Sigma}$ and $\hat{\mu}$ are computed in Eqs. (5)–(6):

$$\hat{\mu} = \mathbb{E}_{\theta \in \mathcal{G}}[\theta] \tag{5}$$

$$\hat{\Sigma} = \mathbb{E}_{\theta \in \mathcal{G}}\left[(\theta - \hat{\mu})(\theta - \hat{\mu})^T\right] \tag{6}$$

This operation increases the number of selected points for the averages computation significantly. However, there is still the risk is of having a variance that is too low. To reduce this risk, the variance of all of the parameters is increased by the same factor (empirically doubled): $\hat{\hat{\Sigma}} = 2\hat{\Sigma}$.

The new vectors are obtained from a classical procedure for multivariate generation:

1. computation of the upper triangular matrix $\mathbf{T}$ with $\mathbf{T}^T \mathbf{T} = \hat{\hat{\Sigma}}$, by Cholesky decomposition;

2. generation of vectors $\tilde{\theta} = \hat{\mu} + \mathbf{T}^T \cdot \varepsilon$, where $\varepsilon$ is a vector with $n_\theta$ independent and normally distributed components with an average of 0 and a variance of 1.

This fourth rule enables us to randomly explore some area of space $\mathcal{E}_\theta$ while implicitly reducing its dimension via the correlations between parameters. This reduces the number of evaluations of the objective function that are needed .

### 3.1.5 Rule 5: recombination

With respect to rule 4, recombination considers that the parameters from a model are not independent. In a hydrological model, they can frequently be grouped in functional blocks (for instance, rapid runoff, base flow, snow dynamics, transfer and so on). A new parameter vector is simply generated by combining blocks of parameters from vectors of the archive $\hat{\mathcal{A}}$. The parameter blocks are specific to the calibrated model and are defined by the user.

## 3.2 Population downsizing

At the end of each generation, the population is kept under a maximum size ($N_{\text{max}}$ sets). This limitation is set for memory reasons (no need to keep poor parameter sets) and to reduce computational time, as the triangulation computation is carried out at each generation.

The population downsizing is adapted from $\varepsilon$-NSGA-II (Reed and Devireddy, 2004) and is performed in three steps (Fig. 4):

1. Pareto ranking – the parameter vectors are sorted according to the ranking order of the Pareto level to which

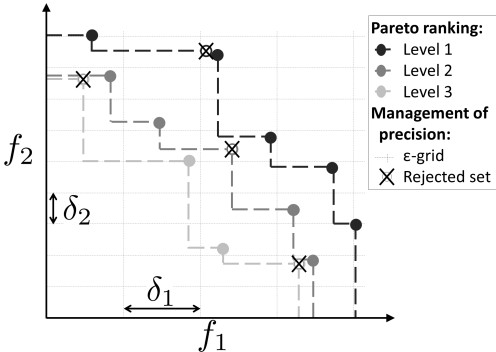

**Figure 4.** Method for population downsizing for a maximization problem with two objectives: Pareto ranking (level 1 is the current approximated Pareto front) and partition of the objective space according to the chosen $\delta_i$ precision (only one vector by hypercube is kept).

they belong. Points from level 1 are non-dominated, points from level 2 are dominated only by points from level 1 and so on.

2. Downsizing according to the chosen precision – the objective space is partitioned by an $n_f$-dimensional grid with the precision $\delta_i$ for each of the $n_f$ objective values. All of the points in the same hypercube are considered to be equifinal with regard to accuracy, and only one point is kept. The selected point is the one that belongs to the lowest Pareto level. When many points are on the lowest level, the selected point is taken at random from among them.

3. Keeping the population size under $N_{\max}$ – if the number of sets is still above $N_{\max}$, only the $N_{\max}$ sets of the lower level are kept.

## 4   Optimization evaluation framework

The aim is to assess the performance of the caRamel algorithm against two other optimizers using various case studies. Two optimizers have been selected for the comparison: NSGA-II (Deb et al., 2002) and MEAS (Efstratiadis and Koutsoyiannis, 2008). The comparison focuses on different aspects: optimization evolution evaluated by specific metrics and optimization results in the objective space, parameters space and observable space. This section presents the optimizer configuration, the evaluation metrics and the four case studies.

### 4.1   Optimizer configurations

The caRamel algorithm is used in its general form, with a generation of five new parameters sets for each rule by iteration, involving an average of 25 parameter sets by generation.

NSGA-II (Deb et al., 2002) is called by using the nsga2 function from the `mco` "Multiple Criteria Optimization Algorithms and Related Functions" (Mersmann et al., 2014) R package. The arguments are the function to minimize, the input and output dimensions, the parameter bounds, the number of generations, the size of the population, and the values for the crossover, mutation probability and distribution index. Some previous calibration experiments have been conducted to determine the best parameter configurations. NSGA-II has been used with a crossover probability set to 0.5 and mutation probability set to 0.3.

The MEAS algorithm (Efstratiadis and Koutsoyiannis, 2005) combines a performance evaluation procedure based on a Pareto approach and the concept of feasibility, an evolving pattern based on the downhill simplex method, and a simulated annealing strategy, to control randomness during evolution. The algorithm evolution is sensitive to the value of the mutation probability which has been adapted to each case study according to its complexity (5 % for Kursawe and 50 % for the other case studies).

For each optimizer, the end of one optimization is set to a maximum number of model evaluations depending on the case studies. As the algorithms use random functions, 40 optimizations of each test case have been run for each optimizer to obtain representative results. In order to focus on the evolution of the optimization, the initial population is the same for each optimizer (40 initial populations for each case study).

We chose to run an important number of model evaluations and optimizations to get representative results and assess the reproducibility of the optimization. Other benchmark methodology would be conceivable, such as that presented by Tsoukalas et al. (2016) where several test functions and two water resources applications are implemented to compare the surrogate-enhanced evolutionary annealing simplex (SEEAS) algorithm to four other mono-objective optimization algorithms. In this study, two alternative computational budgets (indicated by the maximal number of model evaluations) are considered that impact the parameters of the optimizers.

### 4.2   Optimization metrics

To evaluate the optimizer performance, we chose metrics from the literature. Evaluating optimization techniques experimentally always involves the notion of performance. In the case of multi-objective optimization, the definition of quality is substantially more complex than for single-objective optimization problems, because the optimization goal itself consists of multiple objectives (Zitzler et al., 2000). Riquelme et al. (2015) categorize the metrics to evaluate three main aspects:

– accuracy, which is the closeness of the solutions to the theoretical Pareto front (if known) or relative closeness;

– diversity, which can be described by the spread of the set (range of values covered by the solutions) and the distribution (relative distance among solutions in the set);

– cardinality, which qualifies the number of Pareto-optimal solutions in the set.

To quantify these aspects, we selected three different metrics that are evaluated in the objective space:

1. hypervolume (HV), which is a volume-based index that takes accuracy, diversity and cardinality into account (Zitzler and Thiele, 1999) and computes the volume between the vectors of the estimated Pareto front $\hat{\mathcal{F}}$ and a reference point;

2. generational distance (GD), which is a distance-based accuracy performance index (Van Veldhuizen, 1999, Eq. 7) that is expressed as

$$\mathrm{GD} = \frac{\left(\sum_{i=1}^{n} d_i^2\right)^{1/2}}{n}, \tag{7}$$

where $n$ is the number of vectors in the approximated Pareto front $\hat{\mathcal{F}}$, and $d_i$ is the Euclidean distance between each vector and the nearest member of the reference front;

3. generalized spread (GS), which evaluates the diversity of the set (Zhou et al., 2006; Jiang et al., 2014).

The evaluation of the GS and GD metrics requires us to establish a reference front. For each case study, this reference front is built by evaluating the Pareto front on all of the final optimization results of all optimizers.

## 4.3 Case studies

Four case studies have been designed to have an increasing complexity: case study 1 is an analytical example with a Kursawe test function (Kursawe, 1991); case study (2) is on a pluvial catchment with a GR4J open-source hydrological model (Coron et al., 2017, 2019); case study (3) is on a pluvial catchment with a MORDOR-TS semi-distributed model (Rouhier et al., 2017); and case study (4) is on a snowy catchment, also with a MORDOR-TS model.

### 4.3.1 The Kursawe test function

The objective of a test function is to evaluate some characteristics of optimization algorithms. The final Pareto front has a specific shape (non-convex, asymmetric and discontinuous) with an isolated point that the optimizer has to accurately reproduce. The Kursawe function is a benchmark test for many researchers (Lim et al., 2015). It has three parameters ($x_1$, $x_2$ and $x_3$) and two objectives ($\mathrm{Obj}_1$ and $\mathrm{Obj}_2$) to minimize

(Kursawe, 1991, Eq. 8).

$$\begin{cases} \mathrm{Obj}_1 &= -10 \cdot \left( e^{-0.2\sqrt{x_1^2 + x_2^2}} - e^{-0.2\sqrt{x_2^2 + x_3^2}} \right) \\ \mathrm{Obj}_2 &= |x_1|^{0.8} + 5 \cdot \sin(x_1^3) + |x_2|^{0.8} + 5 \cdot \sin(x_2^3) \\ &\quad + |x_3|^{0.8} + 5 \cdot \sin(x_3^3) \end{cases} \tag{8}$$

The optimizations are run on 50 000 model evaluations. The R script to run the Kursawe function optimization with caRamel is available in Appendix B or as a vignette in the `caRamel` package.

### 4.3.2 Calibration of the GR4J model on a pluvial catchment

The GR4J hydrological model is a widely used global rainfall–runoff model (Perrin et al., 2003) that has been implemented in an open-source R package `airGR` (Coron et al., 2017, 2019). This package contains a data sample from a catchment called "Blue River at Nourlangie Rock" (360 km$^2$, code L0123001), which has a pluvial regime (Fig. 5a). The advantage of using this case study is in having an open-source script with open data.

GR4J has four parameters to calibrate: the production store capacity $X1$, the inter-catchment exchange coefficient $X2$, the routing store capacity $X3$ and the unit hydrograph time constant $X4$.

The calibration is done on the daily time series for the period from 1990 to 1999. The Kling–Gupta efficiency (KGE, Gupta et al., 2009) is frequently used in hydrology. The KGE can be split into three components that reflect the correlation between the simulated and observed values (KGE$_r$), the bias in standard deviation (KGE$_\alpha$) and the bias in volume (KGE$_\beta$). The calibration is carried out on these three components (Eq. 9).

$$\begin{cases} \mathrm{KGE}_r = 1 - \sqrt{(1-r)^2} \\ \mathrm{KGE}_\alpha = 1 - \sqrt{(1-\alpha)^2}, & \text{with } \alpha = \sigma_s/\sigma_o \\ \mathrm{KGE}_\beta = 1 - \sqrt{(1-\beta)^2}, & \text{with } \beta = \mu_s/\mu_o, \end{cases} \tag{9}$$

where $r$ is the linear correlation coefficient between simulated and observed time series, $\sigma_s$ and $\sigma_o$ represent their standard deviations, and $\mu_s$ and $\mu_o$ represent their mean values.

For each component, the optimal value is 1 and the optimization consists of a maximization. At the end of the optimization only the sets with KGE$_\beta > 0$ are considered, as a KGE$_\beta$ with a negative value indicates poor quality for hydrological results. This leads us to exclude a few sets for calibration with NSGA-II and caRamel but not for calibration with MEAS.

The R script to run an optimization of the GR4J model with caRamel is available in Appendix C.

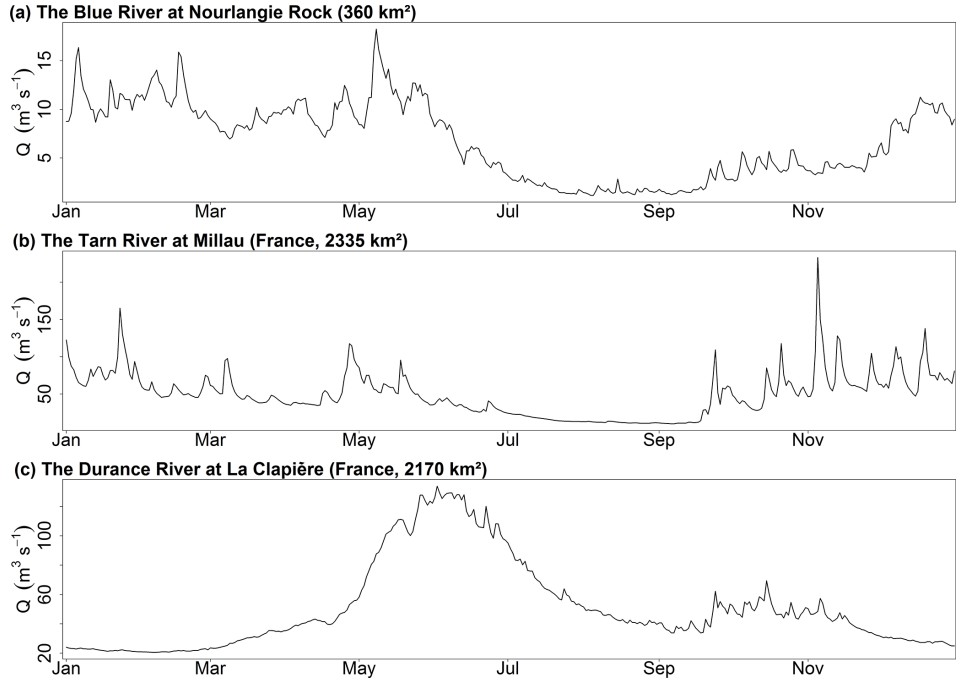

**Figure 5.** Daily discharge regimes at the three catchments studied.

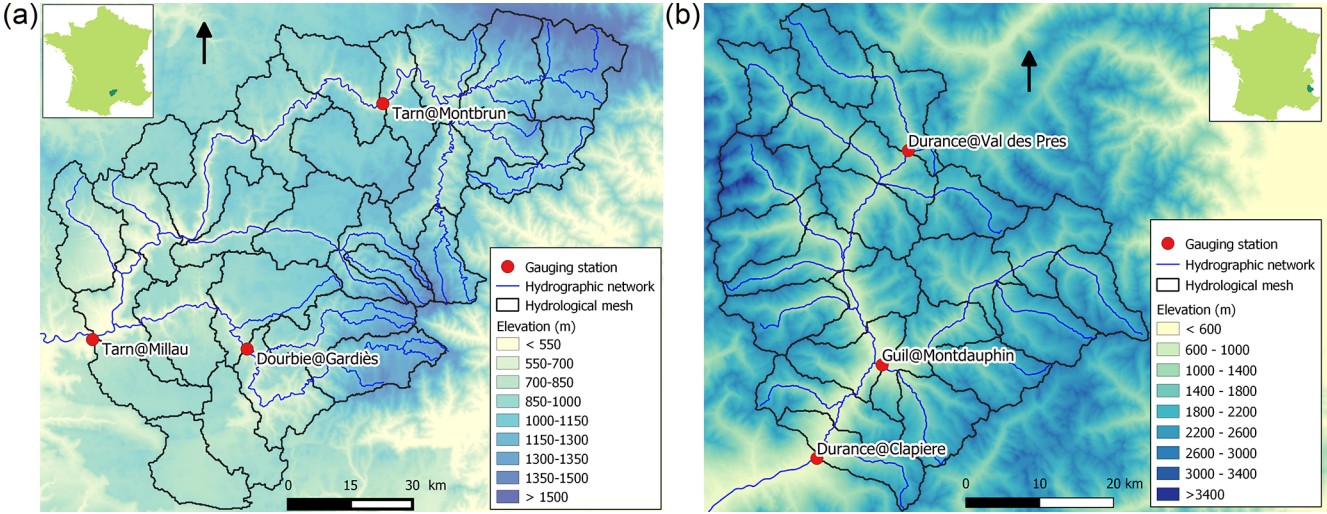

**Figure 6.** Maps of the catchments studied: **(a)** Tarn at Millau (2335 km$^2$) and **(b)** Durance at La Clapière (2170 km$^2$).

### 4.3.3 Calibration of the MORDOR-TS model on two contrasting catchments

The spatially distributed MORDOR-TS rainfall–runoff model (Rouhier et al., 2017) is a spatialized version of the conceptual MORDOR-SD model (Garavaglia et al., 2017) that has been widely used for operational applications at Électricité de France (EDF; the French electric utility company). The catchment is divided into elementary sub-catchments connected according to the hydrographic network which constitutes a hydrological mesh.

This model was implemented at a daily time step for two French catchments with contrasting climates. The Tarn catchment at Millau (Fig. 6a) covers an area of 2335 km$^2$ and has a moderate altitude ranging from 350 to 1600 m. The regime is pluvial, with almost no influence from snow. The Durance at La Clapière catchment (2170 km$^2$, Fig. 6b) is located in the French Alps and has elevations ranging from 800 to about 4000 m. Its hydrological regime is strongly influenced by snow, with a maximum during the melting season in June (Fig. 5c).

**Table 1.** Parameters to calibrate for MORDOR-TS and bounds of variation.

| Parameter | Units | Prior range | Description |
|---|---|---|---|
| cetp | – | [0.7, 1.3] | Potential evapotranspiration multiplicative correction factor |
| cp | – | [0.9, 1.1] | Precipitation multiplicative correction factor |
| gtz | $(°C\,(100\,m)^{-1})$ | [−0.8, −0.4] | Air temperature gradient |
| umax | (mm) | [30, 500] | Maximum capacity of the root zone |
| lmax | (mm) | [30, 500] | Maximum capacity of the hillslope zone |
| zmax | (mm) | [30, 500] | Maximum capacity of the capillarity storage |
| evl | – | [1.5, 4] | Outflow exponent of storage L (intermediate storage) |
| kr | – | [0.1, 0.9] | Runoff coefficient |
| evn | – | [1, 4] | Outflow exponent of storage N (deep storage) |
| lkn | $(mm\,h^{-1})$ | [−8, −1] | Outflow coefficient of storage N |
| kf | $(mm\,°C^{-1}\,d^{-1})$ | [1, 5] | Constant part of melting coefficient |
| kfp | $(mm\,°C^{-1}\,d^{-1})$ | [0, 5] | Variable part of melting coefficient |
| lts | – | [0.7, 1] | Smoothing parameter of snowpack temperature |
| eft | (°C) | [−3, 3] | Additive correction of melting temperature |
| efp | (°C) | [−3, 3] | Additive correction of rain/snow partition temperature |
| cel | $(km\,h^{-1})$ | [0.1, 10] | Wave celerity |
| dif | $(m^2\,s^{-1})$ | [10, 5000] | Wave diffusion |

The hydrological meshes have been built with an average cell area of $100\,km^2$, meaning that 28 cells are needed for the Tarn catchment and 22 cells for the Durance catchment.

MORDOR-TS has 22 free parameters in its comprehensive formulation. For the Tarn case study, a simplified formulation is adopted with 12 free parameters to calibrate in order to describe the functioning of conceptual reservoirs, evapotranspiration correction and wave celerity (Table 1). For the Durance catchment, parametrization of the snow module of MORDOR-TS is more complex, and 16 parameters are to be calibrated for the hydrological model. The parameter distribution is uniform for the two case studies, which means that the same set of parameters applies to all cells. Calibration is conducted over 10 years (1 January 1991–31 December 2000) based on three objectives that have to be maximized.

For the Tarn catchment, the calibration is based on the Nash–Sutcliffe efficiencies (NSE; Nash and Sutcliffe, 1970) at three gauging stations: the catchment outlet (Tarn at Millau) and two interior points (Tarn at Montbrun and Dourbie at Gardiès). For the Durance catchment, the Kling–Gupta efficiency (KGE; Gupta et al., 2009) is computed at three gauging stations: the catchment outlet (Durance at La Clapière) and two interior points (Durance at Val-des-Prés and Guil at Mont-Dauphin). The theoretical optimum is the point (1, 1, 1) in the objective space.

## 5 Results of calibration evaluations

Four aspects are considered with respect to the results of the case studies: the shape of the final Pareto fronts, the dynamics of the optimizations, the distribution of the calibrated parameters and the consequences of the latter on simulated discharges for the hydrological case studies. To illustrate the re-

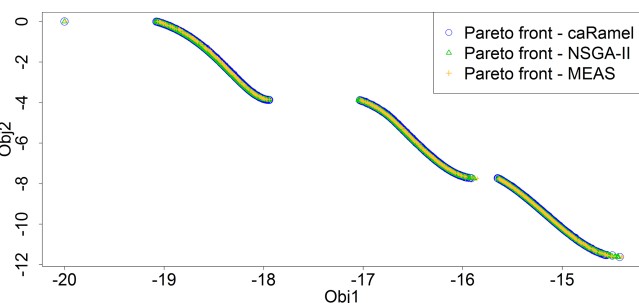

**Figure 7.** Pareto front after 50 000 model evaluations with caRamel (1183 points), NSGA-II (1780 points) or MEAS (687 points) for the Kursawe test function.

sults on the simulated discharges, a "best compromise" set has been selected regarding the distance to point (1,1,1) in the objective space for each hydrological case study.

### 5.1 Final Pareto front

First of all, it is important to accurately reproduce the disconnected Pareto front for the Kursawe test function, and this is the case for all of the optimizers (Fig. 7) with no noticeable differences between the solutions. This confirms the effectiveness of three different algorithms on a low-dimension research benchmark for the multi-objective optimization.

Concerning the three hydrological case studies, the solutions of the Pareto fronts look quite similar for caRamel and NSGA-II and more narrow with MEAS (Fig. 8). The number of sets for the Pareto front changes depending on the case, and there is no rank for the optimizers. For the Blue River study, there are 1172 sets with caRamel, 878 sets with NSGA-II and 268 points with MEAS; there are 1457, 789

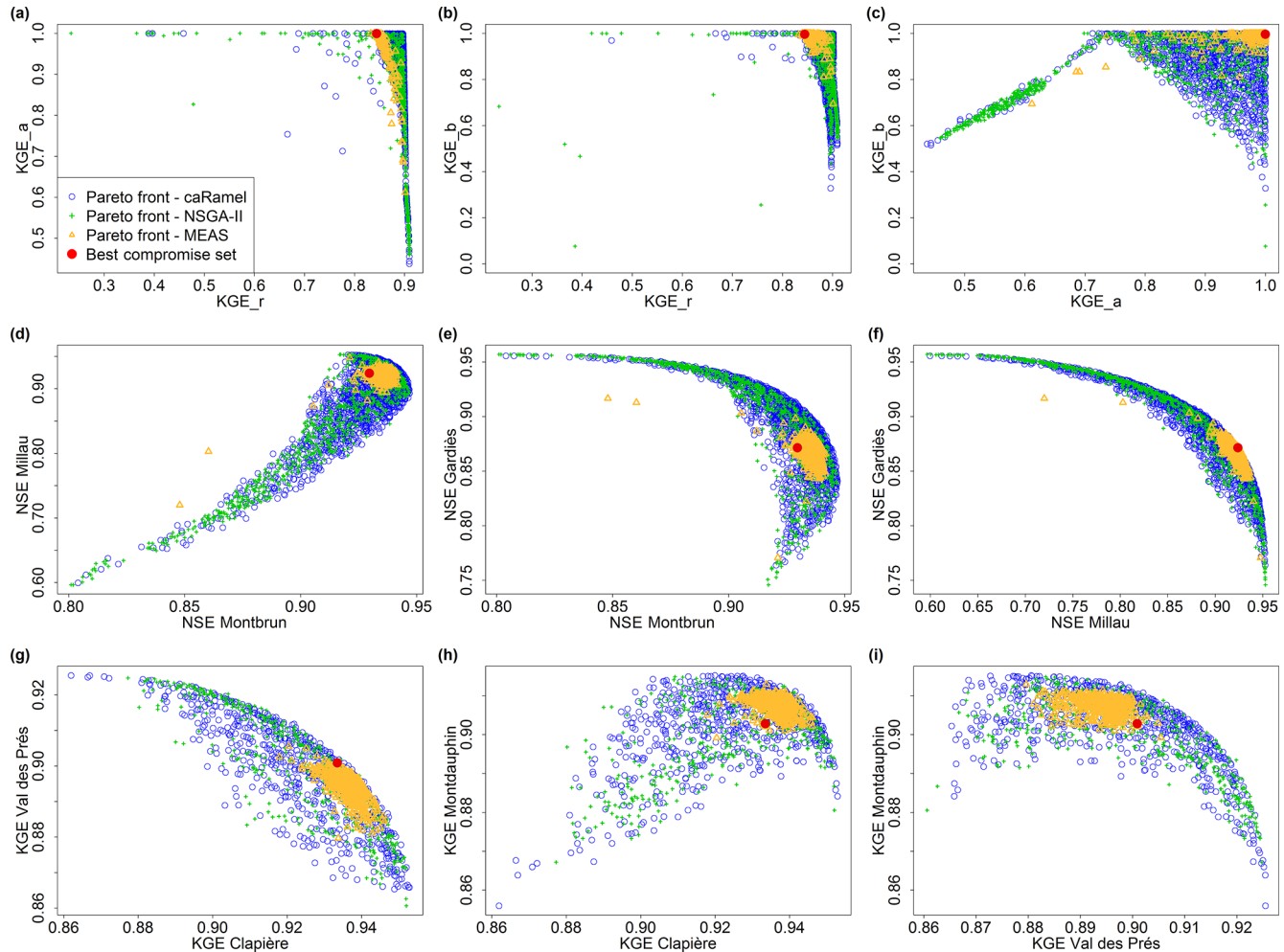

**Figure 8.** Pareto fronts over 40 optimizations with the caRamel, NSGA-II and MEAS optimizers for each hydrological case study: Blue River with GR4J **(a–c)**, Tarn with MORDOR-TS **(d–f)** and Durance with MORDOR-TS **(g–i)**. The red point represents a "best compromise" set that is used to illustrate model results.

and 1882 sets for the Tarn study, and 708, 408 and 525 sets for the Durance study with caRamel, NSGA-II and MEAS respectively. The differences between Pareto fronts are not a priori in favour of a single MEAS-based algorithm. They are given for a limited number of cases which are not necessarily representative of a general behaviour.

## 5.2 Dynamics of the optimizations

Figure 9 summarizes the dynamics of the optimizations for the four case studies.

The caRamel algorithm converges more quickly for accuracy (the HV and GD metrics in most cases). The caRamel's dynamics is closer to NSGA-II's dynamics than to MEAS's dynamics, as they have almost the same final values for the three metrics. This confirms the distinctive behaviour between the two classes of algorithms.

With respect to the diversity criteria, GS dynamics is different for the Kursawe test case than for the hydrological case studies. For the Kursawe test case, the optimal final front has a spread, so all optimizers give the same results. For the hydrological cases, the optimal solution is a point (1,1,1); thus, the Pareto front may get smaller with the optimization. NSGA-II and caRamel look alike, as they generate more diversity than MEAS (GS final values). On average, caRamel gives better values than NSGA-II for the three real cases.

Finally, the envelopes over 40 optimizations are comparable for the three optimizers, which means that reproducibility is always obtained but with different regularities depending on the case or the optimizer without any notable feature. In some cases, a smoother statistical GS convergence would have implied more optimizations.

## 5.3 Parameter distribution

Figure 10 displays the distribution of parameters from the three case studies.

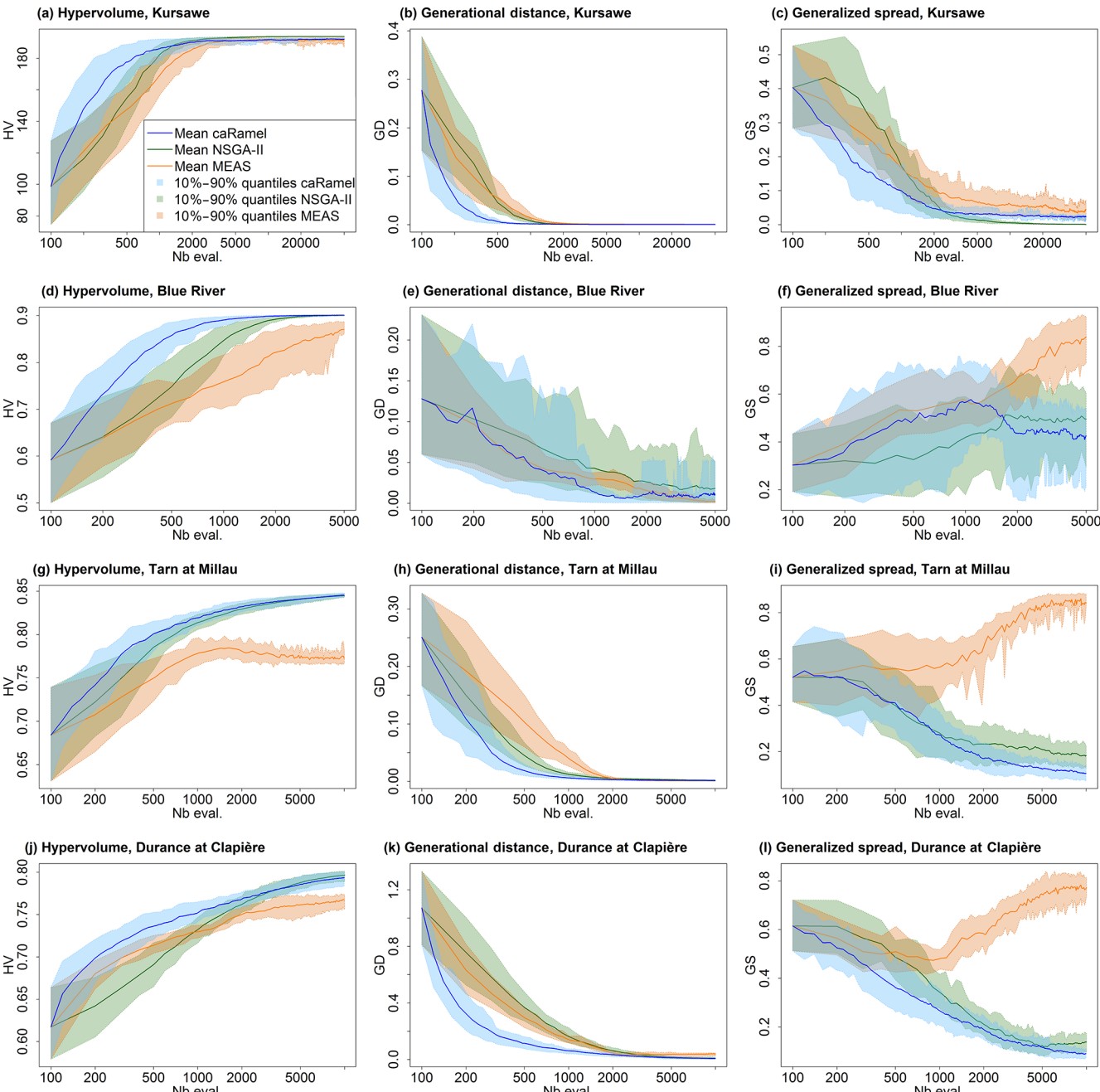

**Figure 9.** Metrics' evolution over 40 optimizations with caRamel, NSGA-II and MEAS, showing the mean evolution and 10 %–90 % quantiles of the metrics with respect to the number of model evaluations: **(a–c)** metrics for the Kursawe test function; **(d–f)** metrics for the GR4J calibration of at Nourlangie Rock; **(g–i)** metrics for the MORDOR-TS calibration of Tarn at Millau; **(j–l)** metrics for the MORDOR-TS calibration of Durance at La Clapière.

In the parameter space, the optimizers provide very similar results that explore the equifinality of the model, meaning that different parameter sets show similar performance (Fig. 10). Some parameters (such as kr or lkn) may have optimized values on the whole range defined by the bounds, whereas other parameters are better constrained ($X4$ and cel).

These constitute a family of sets that are optimal with regard to the chosen objectives.

The difference in the diversity of the final sets is also visible in the parameter distributions. Distributions are quite similar for caRamel and NSGA-II but are much narrower for MEAS. This once again confirms the different behaviour of

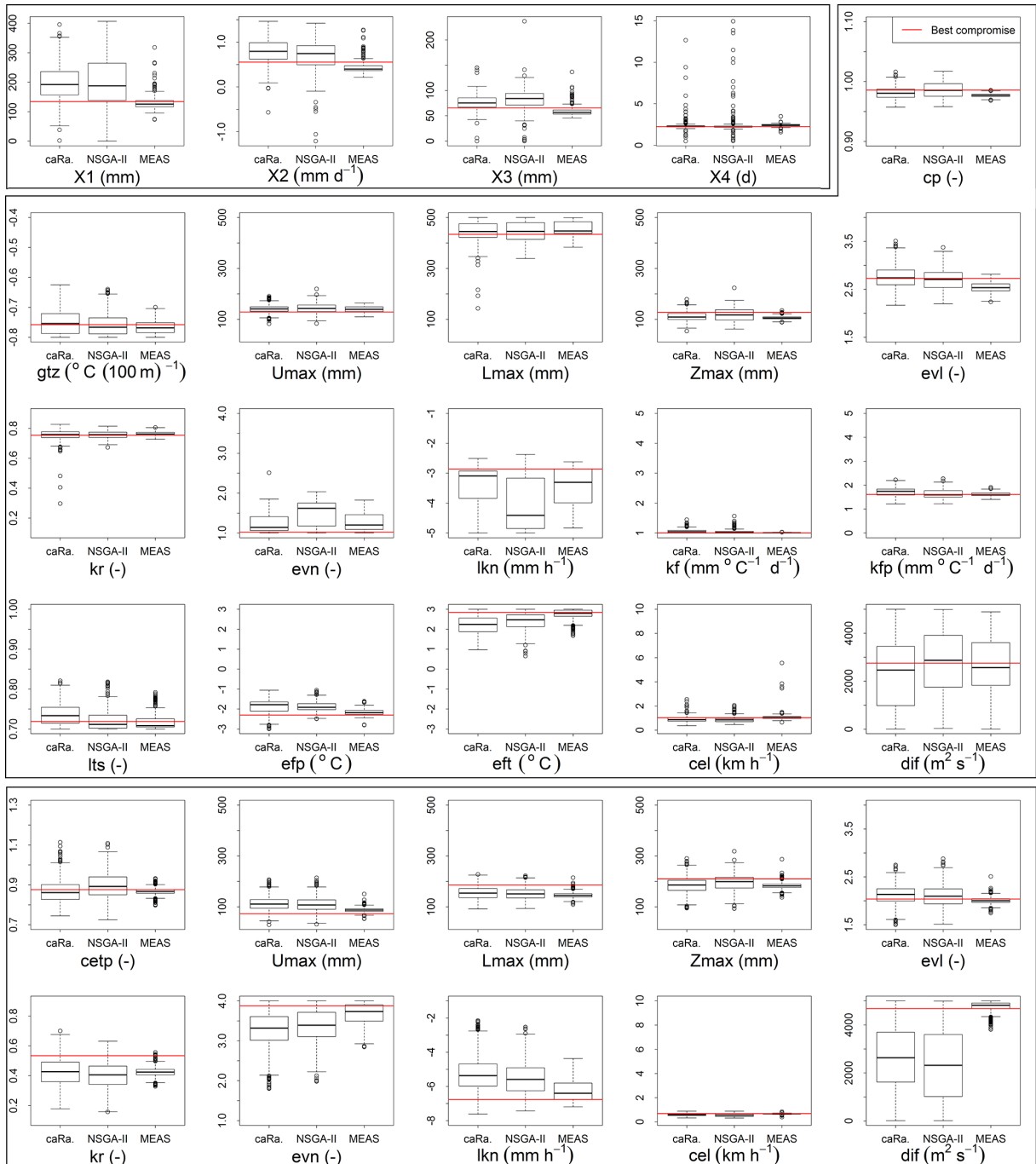

**Figure 10.** Calibrated parameter distributions for the sets on the Pareto front (*y* limits are the calibration bounds, except for *X*1 to *X*4) with caRamel, NSGA-II and MEAS for the three hydrological case studies: Blue River (first block of four parameters), Durance River (second block of 16 parameters) and Tarn River (third block of 10 parameters). Parameter values from the "best compromise" set are displayed in red. TS3

MEAS, with weaker general performance for the cases studied here.

## 5.4 Impact on model results

The consequences with respect to the simulated discharges are displayed in Fig. 11. The envelopes with NSGA-II and caRamel are quite similar, whereas the envelope with MEAS

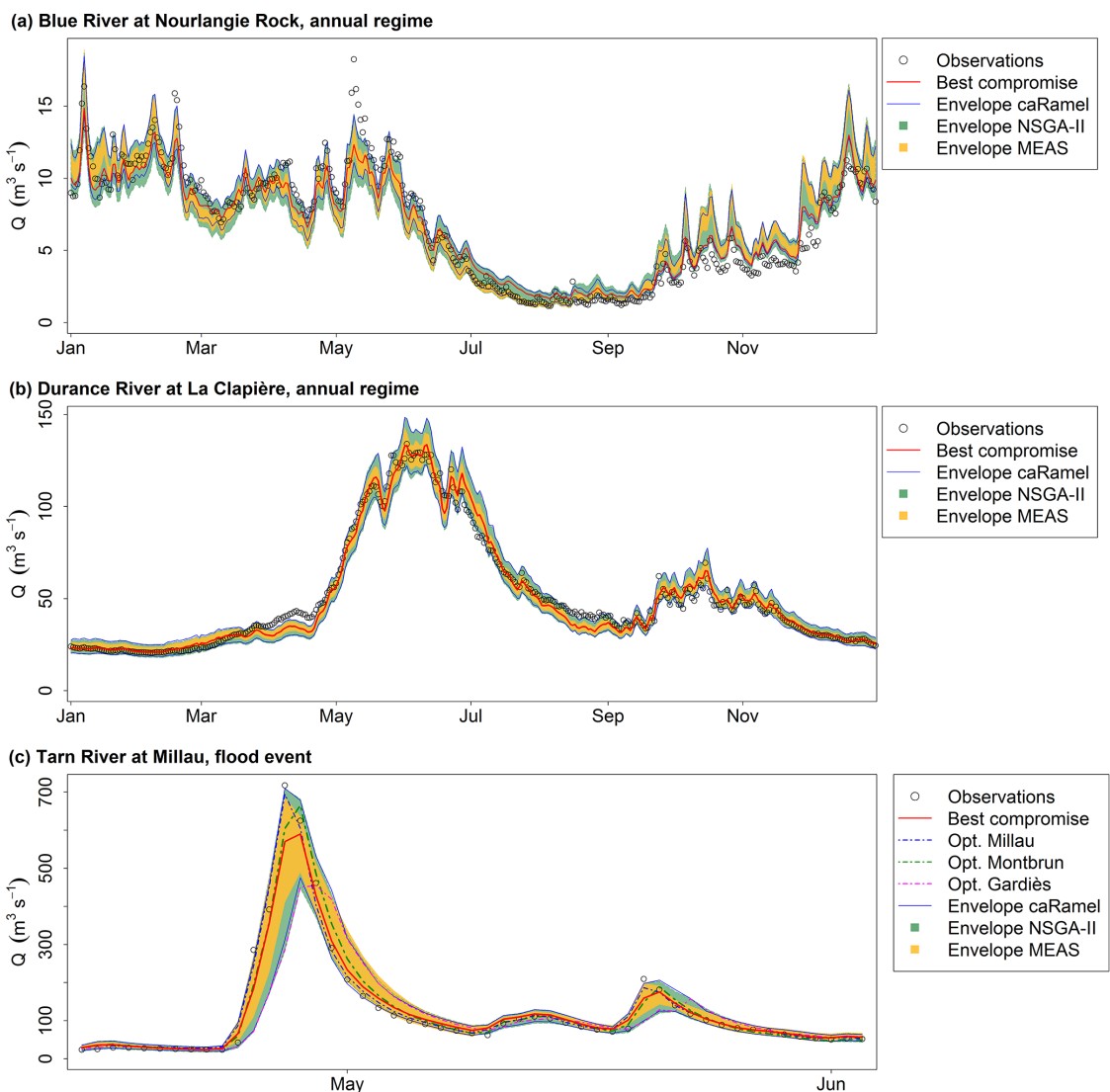

**Figure 11.** Observed and simulated discharges for the three case studies. "Observations" refer to the observed discharges, "Best compromise" refers to the best compromise simulated discharges and "Envelope" refers to the simulated discharges envelope using all parameter sets on the Pareto front (over 40 optimizations) with caRamel, NSGA-II and MEAS. **(a)** Daily runoff regime of Blue River at Nourlangie Rock (1990–1999); **(b)** daily runoff regime of Durance at La Clapière (1991–2000); **(c)** flood event for the Tarn River at Millau (14 April 1993–3 June 1993).

is narrower, as expected. This confirms that caRamel and NSGA-II generate more diversity on their Pareto front. The red line represents the simulated discharges with the best compromise set and fits quite well with the observed discharges. Multi-objective calibration allows for a range of variation of the calibrated discharges around the best compromise simulation.

Figure 11c represents a flood event on the Tarn River at Millau. The observed discharge points are in the envelope of simulation. The best compromise simulation does not accurately reproduce the flood peak. The figure also displays the simulated discharges obtained by optimizing parameters on

the three gauging stations separately, and the simulation with the set that optimizes the NSE at Millau fits better with the observed points.

## 6 Conclusions

The caRamel function is an optimization algorithm for multi-objective calibration, and it results in a family of parameter sets that are Pareto-optimal with regard to the different objectives. The algorithm is a hybrid of the MEAS algorithm (Efstratiadis and Koutsoyiannis, 2005), using the directional search method based on the simplexes of the objective space,

and the $\varepsilon$-NSGA-II algorithm, using the archiving management of the parameter vectors classified by $\varepsilon$ dominance (Reed and Devireddy, 2004). The combination of stochastic and gradient-like parameter generation rules helps the convergence of optimization while preserving the diversity of the population in both the objective and parameter spaces. Four case studies with increasing complexity have been used to compare caRamel with NSGA-II and MEAS. The results are quite similar between optimizers and show that optimization converges more quickly with caRamel.

An optimization algorithm might be delicate to use because of the choice of input arguments, which are specific to the algorithm and might require some "expert knowledge". The sensitivity to caRamel internal parameters is not presented in this paper, but we have carried out some sensitivity analyses using the Morris method (Morris, 1991) in order to recommend some default values to the user. First, it is recommended that users assign the same weight to each generation rule by indicating the same number of parameter sets to generate. It is advantageous to produce a small number of sets by generation in order to reduce the number of model evaluations and obtain more rapid convergence. By default, five sets are generated for each rule. The size of the initial population should be large enough to have enough variability (at least 50 sets for a complex model). Moreover, as convergence can be sensitive to the randomly chosen initial population, it is recommended that the user run two or three optimizations to assess reproducibility.

Multi-objective optimization may require thousands of evaluations, which can be a limitation for the calibration of time-consuming models. To cope with this issue, parallel computation is implemented in the `caRamel` R package.

Better consideration of equality or inequality constraints, such as the relationship between two parameters, could be an improvement. Another perspective would be the ability of caRamel to deal with discrete parameters.

## Appendix A: The `caRamel` R package

The `caRamel` R package has been designed as an optimization tool for any environmental model, provided that it is possible to evaluate the objective functions in R. The main function, caRamel, is called with the following syntax: caRamel(nobj, nvar, minmax, bounds, func, popsize, archsize, maxrun, prec). Arguments are detailed in Table A1. The main argument of caRamel is the objective function that has to be defined by the user. This enables flexibility, as the user gives all of the necessary information: the number and the definition of all the objectives, the minimization or maximization goal for each objective function, the number of parameters to calibrate and their bounds, and other numerical parameters such as the maximum number of simulations allowed. Additional optional arguments give the following possibilities:

- creation of blocks/subsets of parameters that should be jointly recombined (for example, parameters of a same module);

- choice of parallel or sequential computation;

- continuation of optimization starting from an existing population;

- saving of the population after each generation or only the final one;

- indication of the number of parameter sets produced by generation.

As a result, the function returns a list of six elements:

1. success – a logical, "TRUE" if the optimization process ran with no errors,

2. parameters – a matrix of parameter sets from the Pareto front (dimension: number of sets in the front, number of calibrated parameters),

3. objectives – a matrix of associated objective values (dimension: number of sets in the front, number of objectives),

4. save_crit – a matrix that describes the evolution of the optimization process; for each generation, the first column is the number of model evaluations, and the following columns are the optimum of each objective taken separately (dimension: number of generations, number of objectives $+1$),

5. total_pop – total population (dimension: number of parameters sets, number of calibrated parameters + number of objectives).

6. gpp – the calling period for the third generation rule (independent sampling with a priori parameters variance). It is computed by the algorithm if the user does not fix it.

The R package contains an R vignette that gives optimization examples of the Schaffer function (Schaffer, 1984; two objectives, one parameter) and the Kursawe function (Kursawe, 1991; two objectives, three parameters).

**Table A1.** Arguments of the caRamel() function. Optional arguments are shown in italic font.

| Name | Type | Description |
|---|---|---|
| nobj | Integer, length = 1 | Number of objectives to optimize (at least two) |
| nvar | Integer, length = 1 | Number of parameters to calibrate |
| minmax | Logical, length = nobj | Logical vector that indicates whether each objective should be maximized (TRUE) or minimized (FALSE) |
| bounds | Matrix, nrow = nvar, ncol = 2 | Upper and lower bounds for the variables |
| func | Character, length = 1 | The function to optimize (defined by the user), with VecObj = func($i$), where $i$ is the tested set index in the population matrix (**x**), and VecObj is the vector of objectives for this set. |
| popsize | Integer, length = 1 | Population size for the genetic algorithm |
| archsize | Integer, length = 1 | Size of the Pareto front |
| maxrun | Integer, length = 1 | Maximum number of model runs |
| prec | Double, length = nobj | Desired precision for the objectives (used for downsizing the population) |
| *repart_gene* | *Integer, length = 4* | *Number of new parameter sets for each rule and per generation* |
| *gpp* | *Integer, length = 1* | *Calling period for rule (3)* |
| *blocks* | *List of vector integers* | *Functional groups for parameters* |
| *pop* | *Matrix, nrow = nset, ncol = nvar or nvar+nobj* | *Initial population (used to restart an optimization)* |
| *objnames* | *Character, length = nobj* | *Names of the objectives* |
| *listsave* | *List of characters* | *Names of the listing files (NULL by default: no output)* |
| *write_gen* | *Integer, length = 1* | *If the integer = 1, it save files "pmt" and "obj" at each generation (0 by default)* |
| *carallel* | *Logical, length = 1* | *Run parallel computations (TRUE by default)* |
| *numcores* | *Integer, length = 1* | *Number of cores for the parallel computations (all cores by default)* |
| *funcinit* | *Character, length = 1* | *The function (defined by the user) applied on each node of the cluster for initialization for parallel computation (for example, load of packages or copy of data). Arguments must be cl and numcores.* |
| *graph* | *Logical, length = 1* | *Plot graphical output at each generation (TRUE by default)* |

## Appendix B: Example of R script for Kursawe test function optimization

```
# Kursawe function definition
kursawe <- function(i) {
    Obj1 <- -10 * exp(-0.2 * sqrt(x[i,1]^2 + x[i,2]^2)) - 10 * exp(-0.2 * sqrt(x[i,2]^2 + x[i,3]^2))
    Obj2 <- abs(x[i,1])^0.8 + 5 * sin(x[i,1]^3) + abs(x[i,2])^0.8 + 5 * sin(x[i,2]^3) + abs(x[i,3])^0.8 + 5 * sin(x[i,3]^3)
    return(c(Obj1, Obj2))
}
# Parameters definition and caRamel run
nobj <- 2 ; nvar <- 3 ; bounds <- matrix( c(rep(-5, nvar),rep(5, nvar)), ncol = 2 ) # range [-5, 5]
results <- caRamel (nobj = nobj , nvar = nvar , minmax = c(FALSE, FALSE) , bounds = bounds, func = kursawe, popsize =
100 , archsize = 100, maxrun = 5000, prec = rep(1.e-3,nobj) )
```

## Appendix C: Example of R script for GR4J optimization

```
library(airGR)
library(caRamel)
# loading catchment data #
data(L0123001)
# preparation of the InputsModel object
InputsModel <- CreateInputsModel(FUN_MOD = RunModel_GR4J, DatesR = BasinObs$DatesR, Precip = BasinObs$P,
PotEvap = BasinObs$E)
# run period selection
Ind_Run <- seq(which(format(BasinObs$DatesR, format = "%Y-%m-%d")=="1990-01-01"),
which(format(BasinObs$DatesR, format = "%Y-%m-%d")=="1999-12-31"))
# preparation of the RunOptions object
RunOptions <- CreateRunOptions(FUN_MOD = RunModel_GR4J,InputsModel = InputsModel, IndPeriod_Run = Ind_Run)
# Observation object
Obs <- BasinObs$Qmm[Ind_Run]

# Definition of functions for the optimizer #
# Function for model evaluation #
EvalGR <- function(i){
    # Transformation of the parameter set to real space
    RawParamOptim <- airGR::TransfoParam_GR4J(ParamIn = x[i,],Direction = "TR")
```

```r
    # Simulation given a parameter set
    OutputsModel <- airGR::RunModel_GR4J(InputsModel = InputsModel,RunOptions = RunOptions,Param = RawParamOp-
tim)
    # Evaluation of the 3 components of KGE
    Sim <- OutputsModel$Qsim
    ix <- is.na(Obs + Sim)
    B <- sum(Sim[!ix])/sum(Obs[!ix])
    alpha <- sd(Sim[!ix],na.rm = TRUE)/sd(Obs[!ix],na.rm = TRUE)
    rho <- cor(Obs[!ix],Sim[!ix])
    KGE_3 <- c(rho , alpha, B)
    return(1-sqrt((1-KGE_3)^2))
}

# Function for cluster initialization
InitGR <- function(cl,numcores){
    parLapply( cl, 1:numcores, function(xx)require('airGR'))
    clusterExport(cl=cl, varlist=c("InputsModel", "RunOptions", "Obs"))
}

# Optimization #
# definition of the bounds of parameters (between -9.99 and 9.99)
nobj <- 3
bounds <- matrix(c(rep(-9.99, 4),rep(9.99, 4)), ncol = 2)
# Run
results <- caRamel(nobj = nobj, nvar = 4, minmax = rep(TRUE,nobj), bounds = bounds, func = EvalGR, funcinit = InitGR,
popsize = 100, archsize = 100, maxrun = 5000, objnames = c("KGE_r","KGE_a","KGE_b"), prec = rep(1.e-4,nobj))
```

*Code availability.* The data analysis was performed with the open-source environment R (https://www.r-project.org; R Core Team, 2019). The algorithm is provided as an R package "caRamel", which is available from GitHub (https://doi.org/10.5281/zenodo.3895601 **TS4**; Zaoui and Monteil, 2020) or from CRAN (https://cran.r-project.org/package=caRamel; Le Moine et al., 2019). The Blue River at Nourlangie Rocks case study was run using the airGR package for the GR4J hydrological model and for the data set (available at https://cran.r-project.org/package=airGR; Coron et al., 2019).

*Author contributions.* NLM developed the algorithm in the Scilab platform. FH, FZ and CM adapted the algorithm to an R package and performed the various test cases. CM prepared the paper with contributions from all co-authors.

*Competing interests.* The authors declare that they have no conflict of interest.

*Acknowledgements.* The authors wish to thank the editor and the reviewers of this paper for their constructive suggestions. Special thanks to Andreas Efstratiadis, who gave us MEAS source code, and to Guillaume Thirel, who provided the first script of airGR.

*Review statement.* This paper was edited by Elena Toth and reviewed by Andreas Efstratiadis and Guillaume Thirel.

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

**Remarks from the typesetter**

**TS1** Regarding the question from your email, please note that this change affects not only the title, but – more importantly – the manuscript type. At this stage it is usually not possible to change the manuscript type; however, if you insist on doing so, this would require the editor's approval. In this case, please provide a detailed explanation that can be forwarded to the editor. Please note that this entire process will be available online after publication. Upon approval, we will make the appropriate changes. Thank you for your understanding.