# Peer review of "Multi-objective calibration by combination of stochastic and gradient-like parameter generation rules: the caRamel algorithm"

_Hydrology and Earth System Sciences, 2019_

## Editor Comment (EC1) · Elena Toth (Editor) · 29 Jul 2019

Dear Authors, as I anticipated in the submission phase, I have two main suggestions:

1) adding a comparison not only with NSGA-II, but also with the other optimization model you merge in the Caramel (MEAS), since you are proposing an algorithm that should be an improvement above each one of the previous approaches.

2) the algorithm, in terms of hydrological modelling, is only tested on a single catchment (and in addition, the details of such application are lacking): in order to prove the generality of the improvement allowed by the proposed approach, you should test the

calibration of at least another study catchment.

On the abstract/introduction phrasing: p. 1, l. 13: "caRamel()" why the parentheses? p. 1, l "The comparison with another well-known optimizer (i.e. NSGA-II) confirms the quality of the algorithm" p. 2, ll. 7-14: I would replace with: "CaRamel was initially developed and used for the calibration of hydrological models: Le Moine et al., 2015, Rothfuss et al., 2012, Magand et al., 2014, Monteil et al., 2015 (previously to the R-package release) and Rouhier et al. (2017). The interesting performances of caRamel algorithm in such studies prompted us to describe in detail the algorithm in the present paper, and in particular its use as an R-package, that can be used for any model in the R environment. The user has simply to define a vector-valued function (at least 2 objectives) for the model to calibrate and lower and upper bounds for the calibrated parameters. This paper aims at describing the principles of caRamel algorithm and its use as an R-package, through the analysis of its results when used for the parameterisation of an hydrological model . A comparison with the widely used NSGA-II algorithm (available in the R package, "Multiple Criteria Optimization" MCO, Mersmann et al., 2014), is also presented."

Elena Toth

---

## Referee Comment (RC1) · Guillaume Thirel (Referee) · 1 Aug 2019

This manuscript presents an R package useful for calibrating hydrological models against a multi-objective function. It is submitted as a technical note in HESS and as a consequence remains rather short.

The topic of R packages that are dedicated to hydrology is relevant to HESS, as demonstrated the very recent review paper by Slater et al. (2019) in which I participated. I think that this paper should be mentioned in the manuscript, not because I co-authored it, but because it justifies the interest of such packages to the hydrological community. Indeed, a basic search for the word "package" in the titles or abstracts of HESS papers

does not provide so many results. . .

I understand from discussions between the editor and the authors (not available online) that a first version was submitted to another journal and it was suggested to focus more on the package than on actual comparison of algorithms impacts. That can be true for model-oriented journals, such as EM&S and GMD (see for instance our airGR paper in EM&S, Coron et al., 2017, which was a Short Communication). However in my opinion HESS is different: describing software and providing pieces of codes is very much appreciate (for the sake of reproducibility), but we also need science. Simply describing packages or tools is not the main aim of HESS. Here the science is a new calibration algorithm and its impact. As a consequence, more emphasis should be put on that two points, but the Editor Elena Toth already commented on that and I agree with her. In the end, I'm wondering if this paper will remain a Technical note or become a full research article.

Overall, the paper is good, it is of interest for the HESS readers in my opinion, and provided that the objectives of the paper are modified as mentioned below and above, I am confident that it will be published later on.

Main criticism:

My main criticism regards the fact that the authors submitted a manuscript into an open journal, for presenting an open source software, but they illustrate it with an hydrological model that is not open! While I know that MORDOR is the historical EDF hydrological model and using the MORDOR model with caRamel makes sense for the authors, this in my opinion is much less obvious for the readers. I guess that the aim of the authors is to make readers understand that caramel is a valuable tool for calibrating hydrological models available in R. However, no mention is ever made of the fact that there are hydrological models available in the R environment, justifying the interest of this new package. Indeed, readers cannot manipulate the MORDOR model, they only have brief knowledge of the parameters meaning from this paper and they cannot run

the lines of codes provided by the authors to handle the package. I think that this is a pity, as the authors made a laudable effort to provide an open source software, some vignettes and also some lines of codes. In addition to the fact that the model is not available, no example data are provided. All of this does not prevent from using the package, but I find it damageable to stop so close to total openness. As one of the developers of the airGR R package, as you know, I tested the caRamel algorithm with a simple GR4J model and an example dataset included in the airGR package. It is not difficult to make it work, but from what was provided it is not straightforward (I guess that the fact that I know well my package helped, but for people using caRamel with a package they did not develop it could be more difficult). I provide the lines of codes I wrote at the end of this review. I would therefore suggest to the authors to include a full example, i.e. one that could be reproduced by the readers, with one open (and well-known if possible!) hydrological model and one open dataset. I don't think this is much work and I don't think that would deeply impact the paper, but I feel that would be useful. Of course, as I provide some lines of codes in this review (see the figures 1, 2, 3, as well as the supplement for the R file) and as some authors know well the GR models, airGR would be a straightforward solution, but as highlighted in Slater et al. (2019) (see section 3.5), other well-known hydrological models are openly available in other R packages.

Major remarks:

Introduction: nothing is said about composite criteria (e.g. an average of KGEs). I think it should be mentioned and discussed on the base of a couple of references. That's what I tested in the code I provided and the results in terms of performance are quite similar to the multi-objective calibration with the same objectives. This is also much faster, but I guess that the choice of the convergence threshold of caRamel is important in this example.

Heterogeneous spelling in the text: see for instance "Multi Objective" in the title, "multi-objective" and "multiobjective" in the abstract. Or "modelling is the abstract but "mod-

elling" in L. 2, P. 2. "R package" in the title and "R-package" in L. 8, P. 2. "Pareto-front" in P. 5, L. 14 and "Pareto front" in P. 5, L. 28, "Pareto Front" in P. 9, L. 12. Please try to be consistent all over the manuscript! I did not spot all of them here.

About section 2.1.1, which describes the generation of parameters. I had difficulties to understand the complete functioning of the 5 rules. First, are these 5 rules or in fact 5 steps that are undertaken successively? In addition, a figure is presented for the first two steps, but it is not used efficiently to make things clear for the readers. I have the feeling that this could be improved easily. For example, why not using only 2 parameters instead of 3, as 3D plots are not possible here? That would simplify things. There are also terms in the legend that are not explained, such as "Example" or "Simplexes". Since the figure is quite complex, with many points, triangles, arrow, it is necessary to help us to read it. I would also maybe suggest a concrete example somehow, especially about the evolution of the parameters sets number, as it is stated at L. 20 that it is necessary to reduce the number of sets. Maybe specify that no parameter set is discarded during the 5 rules stuff?

Section 2.1.2: quite similarly to the previous section, I find that Fig. 2 is underexploited / underexplained. Levels 2 and 3 are never referred to either in the text or in the caption.

Section 4.1: Results are presented for the Kursawe function in Fig. 3. Here again some improvement is possible that would help our reading. What is the optimum for the two objectives? Caption: MCO is not mentioned regarding the number of evaluations; is it 1000? Or does that mean that Fig. 3a only concerns caRamel? If so, why? (I guess this is the case from the reading of the text, but then the caption could be improved). We could have a similar graph for mco.

Section 4.2.4: Figure 5 is quite difficult to read, in the sense that there are too many points and the superposition of the two algorithms results does not permit to see the differences. Maybe you should separate the two algorithms in two different graphs for each panel? In addition, we don't know from which algorithm are chosen the red points,

if I'm not wrong.

Regarding the name of the package, I recommend, if you use LaTeX, to use the \texttt font. This is what we did in Slater et al. (2019) and I think it helps the reading a lot. In addition, it helps making the distinction between the package and the main function, which have the exact same name. We also adopted the spelling "R package" instead of "R-package".

Section 3: I would like to have access to the equations of the metrics, as these are not common metrics used by hydrologists.

References:

- Efstratiadis and Koutsoyiannis: this is a conference abstract. What do you think about citing their HSJ paper entitled "One decade of multi-objective calibration approaches in hydrological modelling: A review¿?

- Garavaglia et al. : this paper has been published in 2017, please update.

- Le Moine 2009: I think we need a link towards the pdf of this report.

- Monteil et al.: any paper or accessible report or presentation instead of this EGU abstract?

- Riquelme et al.: Conférence is written in French

Minor typos and miscellaneous minor stuff:

I found a bunch of very minor typos that could easily be dealt with by the authors. As a non-native speaker myself, I hope I'm correct when suggesting these modifications and also that I'm not too picky... I also do not guarantee that I spotted all of them!

Page 1:

L. 12: remove "algorithm" after NSGA-II as the A of NSGA stands for "algorithm"

L. 20: The first sentence of the introduction is the same as the first one of the abstract,

except for the word "calibration" that is replaced with "prescription". I personally prefer "calibration".

L. 22: I would add a comma after "precisely"

L. 23: not 100% sure, but I think that here "statistic" should be spelt "statistics"

L. 24: "well-knowN"

Page 2:

L. 7-8: please revise the format of citations (i.e. add commas around the years). In addition, I think that the citations should be ordered either by alphabetical or by chronological order.

L. 8: as I guess this is the case for all citations, I would rephrase the parenthesis as follows: "all previously to the release of the R package"

L. 9: "of THE caramel algorithm"

L. 11: I would say "simply has" instead of "has simply"

L. 17: as it is an adjective to the word "objective", I think that "low-flow" is more correct. At this line and all other occurrences, I would remove the capital letter to "hydrology".

L. 22: either "one additional objective" or "additional objectiveS". "Constrain", not "constraint".

L. 24: "THE caRamel algorithm belongs to THE. . ."

L. 26: "THE caRamel. . ."

Page 3:

L. 2: I would say "The caRamel algorithm"

L. 16: "fRont"

L. 20: "each generation": does each of the 5 rules counts for one generation?

Page 4:

L. 6: "this syntax". In addition, please remove the space after "caRamel" and before the parenthesis.

L. 9-10 "the minimization or maximization goal": from the way it is written, I was expecting a value. Actually, it is a Boolean used for saying if this is a maximization (minmax = TRUE) or not. I would suggest rephrasing. In addition, I think that the name of the argument, minmax, is not well chosen. minmax is ok if the choices are "min" and "max", but not if the choices are "TRUE" or "FALSE".

L. 15: I would say "choice of a parallel or sequential computation"

L. 18: not clear. Which kind of managing?

L. 19: there are six elements when I tried it, but the last one (gpp) is not explained in the package.

L. 23: I would say "of objective values", as otherwise we don't know if these are names or values.

Page 5:

L. 6: "an R vignette that gives"

L. 14, 15, 17: I would add a comma after accuracy, diversity, cardinality.

L. 20: please add ", which" after "(GS)"

L. 21: please add a comma after "(HV)" and an hyphen in "volume-based"

L. 24: I would rather say "the R package \texttt{mco} ("Multiple Criteria Optimization", Mersmann et al., 2014)". Please also make sure in the whole manuscript that the package name is written as mco, not MCO.

Page 6:

L. 2: "with THE Kursawe...". In addition, I think it is necessary to introduce a bit what this function, is, what is its aim. I'm not familiar with it and I guess that some other hydrologists also are not.

L. 4: "THE population size"

L. 5 and 8: "THE caRamel ..."

L. 10: double typo: "show that caRamel"

Page 7:

L. 11: maybe specify on which spatial unit the model is semi-distributed.

Page 8:

L. 2 then is the snow part of the model active?

L. 8: define the KGE acronym.

L. 10: maybe replace "is the result of" with "reflects"

L. 16: here and everywhere else, I would write "parameter sets" with no S at parameter.

Page 9:

L. 4: "In the mco..."

L. 17: "optimizerS" and "getS"

L. 18-19: what is the meaning of reproducible? I don't get it.

Page 10:

L. 4: "provide", not "provides"

L. 5: "give", not "giving"

L. 6: I quite disagree regarding the "cel" parameter as especially for mco, the spread seems quite large on Fig. 7.

L. 13: any idea about how many parameter sets that represents?

Page 11:

L. 10: "in an R. . ."

L. 11: "THE result"

L. 14: "from THE mco R package. . ."

L. 18: "provided that"

L. 21: "as an R package"

Appendix A and B:

In the code, some spaces are randomly put before or after commas. That could be cleant.

Table 1: caption: "of THE caRamel. . ."

I have a couple of comments regarding the descripotion of the main function of the package:

- nvar: why not using the word parameter instead of variable? See the description of argument bounds, which uses the name "parameter"

- minmax: somehow it should appear that several values are possible (e.g. "whether the objectives are. . .")

- func: "the R objective function"

- objnames: "nameS"

- write_gen: that would be much more logical to have a logical here instead of an

integer

- func and funcinit: these two arguments do not take characters but functions. If we enter "kursawe" it does not work, we have to put the name of the function. See FUN argument of tapply for instance.

Table 2: cetp is a "multiplicative" correction factor

References:

Coron, L., Thirel, G., Delaigue, O., Perrin, C., Andréassian, V., 2017: The Suite of Lumped GR Hydrological Models in an R package, Environmental Modelling & Software, 94, 166-171, DOI: 10.1016/j.envsoft.2017.05.002.

Slater, L. J., Thirel, G., Harrigan, S., Delaigue, O., Hurley, A., Khouakhi, A., Prosdocimi, I., Vitolo, C., and Smith, K.: Using R in hydrology: a review of recent developments and future directions, Hydrol. Earth Syst. Sci., 23, 2939-2963, https://doi.org/10.5194/hess-23-2939-2019, 2019.

Please also note the supplement to this comment:
https://www.hydrol-earth-syst-sci-discuss.net/hess-2019-259/hess-2019-259-RC1-supplement.zip

```
################### test GR ###################
rm(list = ls())
library(airGR)
library(caRamel)

**loading catchment data**
data(L0123001)

**preparation of the InputsModel object**
InputsModel <- CreateInputsModel(FUN_MOD = RunModel_GR4J, DatesR = BasinObs$DatesR,
                                 Precip = BasinObs$P, PotEvap = BasinObs$E)

**run period selection**
Ind_Run <- seq(which(format(BasinObs$DatesR, format = "%Y-%m-%d")=="1990-01-01"),
               which(format(BasinObs$DatesR, format = "%Y-%m-%d")=="1999-12-31"))

**preparation of the RunOptions object**
RunOptions <- CreateRunOptions(FUN_MOD = RunModel_GR4J,
                               InputsModel = InputsModel, IndPeriod_Run = Ind_Run)

**single efficiency criterion: KGE**
InputsCrit <- CreateInputsCrit(FUN_CRIT = ErrorCrit_KGE,
                               InputsModel = InputsModel, RunOptions = RunOptions,
                               Obs = list(BasinObs$Qmm[Ind_Run]),
                               VarObs = "Q", transfo = "",
                               Weights = NULL)
**single efficiency criterion: KGE sqrt**
InputsCrit2 <- CreateInputsCrit(FUN_CRIT = ErrorCrit_KGE,
                                InputsModel = InputsModel, RunOptions = RunOptions,
                                Obs = list(BasinObs$Qmm[Ind_Run]),
                                VarObs = "Q", transfo = "sqrt",
                                Weights = NULL)

InitGR<-function(cl,numcores){
  parLapply( cl, 1:numcores, function(xx){require('airGR')})
  clusterExport(cl=cl, varlist=c("InputsModel", "RunOptions", "InputsCrit", "InputsCrit2"))
}

EvalGR <- function(i){## Transformation of the parameter set to real space
  RawParamOptim <- airGR::TransfoParam_GR4J(ParamIn = x[i,],
                                            Direction = "TR")
  ## Simulation given a parameter set
  OutputsModel <- airGR::RunModel_GR4J(InputsModel = InputsModel,
                                       RunOptions = RunOptions,
                                       Param = RawParamOptim)
  ## Computation of the value of the performance criteria
  OutputsCrit <- airGR::ErrorCrit_KGE(InputsCrit = InputsCrit,
                                      OutputsModel = OutputsModel,
                                      verbose = FALSE)
  ## Computation of the value of the performance criteria
  OutputsCrit2 <- airGR::ErrorCrit_KGE(InputsCrit = InputsCrit2,
                                       OutputsModel = OutputsModel,
                                       verbose = FALSE)
  return(c(OutputsCrit$CritValue, OutputsCrit2$CritValue) )
}

**definition of the bounds of parameters (between -9.99 and 9.99)**
**the parameter values will be transformed to actual parameter values in EvalGR**
bounds <- matrix(c(rep(-9.99, 4),rep(9.99, 4)), ncol = 2)
```

**Fig. 1.**

```
results  <-  caRamel(nobj = 2, nvar = 4, minmax = rep(TRUE,2), bounds = bounds, func = EvalGR,
                     funcinit = InitGR, popsize = 100, archsize = 100, maxrun = 15000,
                     prec = rep(1.e-4,2) )

Param_temp <- results$parameters[round(dim(results$parameters)[1]/2),] #choosing a kind of median parameter set
Param <- airGR::TransfoParam_GR4J(ParamIn = Param_temp, Direction = "TR") # transforming back to actual parameter values
**Simulation given the optimised parameter set**
OutputsModel <- airGR::RunModel_GR4J(InputsModel = InputsModel,
                                      RunOptions = RunOptions,
                                      Param = Param)
**Computation of the value of the performance criteria**
OutputsCrit <- airGR::ErrorCrit_KGE(InputsCrit = InputsCrit,
                                     OutputsModel = OutputsModel,
                                     verbose = FALSE)## Computation of the value of the performance criteria
OutputsCrit2 <- airGR::ErrorCrit_KGE(InputsCrit = InputsCrit2,
                                      OutputsModel = OutputsModel,
                                      verbose = FALSE)
print(c(OutputsCrit$CritValue, OutputsCrit2$CritValue))

############### comparison the airGR calibration (Calibration_Michel) and a composite OF ######

rm(list = ls())
library(airGR)

**loading catchment data**
data(L0123001)

**preparation of the InputsModel object**
InputsModel <- CreateInputsModel(FUN_MOD = RunModel_GR4J, DatesR = BasinObs$DatesR,
                                 Precip = BasinObs$P, PotEvap = BasinObs$E)

**run period selection**
Ind_Run <- seq(which(format(BasinObs$DatesR, format = "%Y-%m-%d")=="1990-01-01"),
               which(format(BasinObs$DatesR, format = "%Y-%m-%d")=="1999-12-31"))

**preparation of the RunOptions object**
RunOptions <- CreateRunOptions(FUN_MOD = RunModel_GR4J,
                               InputsModel = InputsModel, IndPeriod_Run = Ind_Run)

**efficiency composite criterion: KGE mixing**
**both raw and sqrt-transformed flows**
InputsCritCompo <- CreateInputsCrit(FUN_CRIT = list(ErrorCrit_KGE, ErrorCrit_KGE),
                                    InputsModel = InputsModel, RunOptions = RunOptions,
                                    Obs = list(BasinObs$Qmm[Ind_Run], BasinObs$Qmm[Ind_Run]),
                                    VarObs = list("Q", "Q"), transfo = list("", "sqrt"),
                                    Weights = list(0.5, 0.5))

**preparation of CalibOptions object**
CalibOptions <- CreateCalibOptions(FUN_MOD = RunModel_GR4J, FUN_CALIB = Calibration_Michel)

**calibration**
start time <- Sys.time()
```

**Fig. 2.**

```
OutputsCalib <- Calibration_Michel(InputsModel = InputsModel, RunOptions = RunOptions,
                                   InputsCrit = InputsCritCompo, CalibOptions = CalibOptions,
                                   FUN_MOD = RunModel_GR4J)
end_time <- Sys.time()
end_time - start_time

**simulation**
Param <- OutputsCalib$ParamFinalR
OutputsModel <- RunModel_GR4J(InputsModel = InputsModel,
                              RunOptions = RunOptions, Param = Param)

**Computation of the value of the performance criteria**
**single efficiency criterion: KGE**
InputsCrit <- CreateInputsCrit(FUN_CRIT = ErrorCrit_KGE,
                               InputsModel = InputsModel, RunOptions = RunOptions,
                               Obs = list(BasinObs$Qmm[Ind_Run]),
                               VarObs = "Q", transfo = "",
                               Weights = NULL)
**single efficiency criterion: KGE sqrt**
InputsCrit2 <- CreateInputsCrit(FUN_CRIT = ErrorCrit_KGE,
                                InputsModel = InputsModel, RunOptions = RunOptions,
                                Obs = list(BasinObs$Qmm[Ind_Run]),
                                VarObs = "Q", transfo = "sqrt",
                                Weights = NULL)
OutputsCrit <- airGR::ErrorCrit_KGE(InputsCrit = InputsCrit,
                                    OutputsModel = OutputsModel,
                                    verbose = FALSE)## Computation of the value of the performance criteria
OutputsCrit2 <- airGR::ErrorCrit_KGE(InputsCrit = InputsCrit2,
                                     OutputsModel = OutputsModel,
                                     verbose = FALSE)
print(c(OutputsCrit$CritValue, OutputsCrit2$CritValue))
```

**Fig. 3.**

---

## Referee Comment (RC2) · Andreas Efstratiadis (Referee) · 2 Aug 2019

General comments and critique

The article presents a recently developed multiobjective calibration algorithm and its implementation in R environment, under the title CaRamel. Its background lies upon two different schemes, i.e. the Multiobjective Evolutionary Annealing Simplex method (MEAS) and the Nondominated Sorting Genetic Algorithm II ($\varepsilon$-NSGA-II). The algorithm is tested against a mathematical problem and a real-world hydrological calibration problem, exhibiting good performance in terms of several metrics.

My overall opinion about this article is positive, yet in its current form cannot stand neither as a technical note not as research paper. Actually, it rather resembles to an extended abstract of a clearly hard and long research, which may be useful as a brief documentation for the R community, but is not suitable for a top hydrological journal such as HESS. First of all, the authors have to decide the orientation and the objectives of this article. There are several alternatives, i.e. (a) a state-of-the-art discussion of the multiobjective calibration problem in hydrology; (b) a comprehensive description and justification of the algorithm and its technological advances, accompanied by extended tests of its performance against problems of varying complexity and against other well-established methodologies, and (c) a more synoptic description of the algorithm, with emphasis to its application to few (not only one) representative hydrological calibration problems of varying difficulty, to be presented and discussed in detail.

In this context my recommendation is for a major revision, towards the formulation of a substantially enhanced research paper (not a technical note).

Specific comments

Page 1, lines 13-14: "The main function of the package, caRamel(), requires to define a multi-objective calibration function as well as bounds on the variation of the underlying parameters to optimize". Too obvious technical detail to be referred in the abstract.

Page 1, lines 24-25: "... it is well-know that errors in a simulated discharge time series are not normally distributed, and do not have constant variance and autocorrelation." This statement is true (a reference would be helpful, e.g. Sorooshian and Dracup, 1980), but is not so much evidently linked with the need for multiobjective calibration. Actually, the multiobjective approach in hydrological modelling covers much more cases, including fitting to multivariable and multisite data, as well as soft information (cf., Madsen, 2003; Efstratiadis Koutsoyiannis, 2010).

Page 1, lines 28-29: "Evolutionary algorithms have become widely used to explore the Pareto-optimal front in multi-objective optimization problems that are too complex

to be solved by descent methods". Do they exist descent methods for multiobjective optimization? Maybe you refer to classical aggregation approaches (e.g. weighting of criteria) that have to be solved multiple times with different weighting values, in contrast to evolutionary approaches that only require "a single optimization run", as correctly mentioned just after (page 2, line 2).

Page 2, lines 5-6: "The caRamel optimizer has been developed to meet the need of an automatic calibration procedure that delivers not only one but a family of parameters sets that are optimal regarding a multi-objective target". There do exist many algorithms covering this general objective. Is there any specific objective for the development of caRamel? Which shortcomings of the existing algorithms have you detected before deciding building a new method?

Page 2, line 17: Terms "flood objective" and "low flow objective" are unclear (at least for a non-expert). Page 2, lines 17-19: "Multi-objective calibration is also a way to add some constraints to an underconstrained problem when many parameters have to be quantified. This can help to reduce the equifinality of parameters sets". More discussion should be made here (for 30 years, equifinality remains a hot topic in hydrology), including some representative references, e.g. Her and Seong (2018).

Page 2, lines 20-21: "Equifinality may be caused by the model structure, when two sets of parameters give similar results. Another kind of equifinality is related to the calibration objectives, when two different model results give similar objective values." Term "result" is unclear – probably you refer to the model outputs, by means of response time series. In this respect, two different parameter sets, except if they are very close, cannot provide the same outputs (i.e., similar individual values), they can provide outputs with similar statistical characteristics, and thus similar performance metrics, as correctly stated in the second phrase.

Page 2, line 28: Please, also cite the more detailed and peer-reviewed paper by Efstratiadis and Koutsoyiannis (2008), published as book chapter.

[Figure]

Page 3, section 2.1.1 (Generation rules): The description of the algorithm is very poor and only provides a very general idea about the generation mechanisms. How are these rules associated with the ones used in MEAS? I see quite many differences and very interesting ideas implemented here, but the text is too short to allow understanding and evaluating the methodology (and its potential novelties). Figure 1 is also little helpful; for instance, green and blue points, indicating new sets, are missing, although they are referred in the legend.

Page 3, line 25: Why you keep points of the lower level? Aren't they dominated by points of the upper one?

Pages 4-5, section 2.2 (The caRamel R package): This section is very technical and not so much relevant with the broader philosophy of HESS.

Page 5, line 15: "The diversity which can be described with two aspects: the spread of the set ..." Diversity may refer both to the parameter and the objective space. Which of the two sets are used here?

Page 5, line 25: Please, cite Deb et al. (2002) who developed NSGA-II.

Page 6, lines 12-13: "Comparison with MCO (NSGA-II only) shows that the use of MEAS makes the optimization process converge more rapidly but with a lower diversity". Can you explain the reason of this behavior? Is this an inherent drawback of caRamel, or is due to the algorithmic inputs used in this experiment? As shown in Table 1 (and similarly to all hybrid optimization schemes), caRamel uses quite a large number of input arguments that need manual tuning. Did you run the algorithm by testing alternative set-ups? Do you have any recommendations for the users, regarding the selection of these inputs?

Page 7, section 4.2 (Hydrological modeling): Your case study does not allow extracting safe conclusions about the performance of your method and its comparison against NSGA II. The key reason is that the use of a single overall metric, i.e. KGE, ensures

almost perfect fitting to observations (KGE = 95

Page 8, line 8: Please, better explain criteria (2) and (3) and the associated signatures. Have been these criteria used elsewhere? If yes, please also provide the associated references.

Page 9, lines 3-4: "MCO has been used with crossover probability set to 0.5 and mutation probability to 0.3". Have you made any preliminary tests before selective these values? Which are the values applied to the input arguments of caRamel?

Page 9, Figure 5: I find your figure a little bit misleading. In the vertical axis, the spread of solutions is very small, and within the anticipated range of uncertainty induced in any hydrological calibration exercise. For instance, the lower value of KGEamd is 0.83, while the higher is 0.86. From my point-of-view, such differences do not make sense in the real world.

Page 10, line 12: How did you selected the best compromise parameter set? What do you mean by term "observed set"?

Page 11, section 5 (Conclusions): This section is poorly developed. It has to be written from scratch, to highlight the advantages and weaknesses of the methodology and also discuss ideas for future research.

Minor editorial comments

Page 2, line 10: In which of the aforementioned papers do you describe the algorithm? It is not clear here.

Page 2, line 17: Term "Hydrology" should not start with capital.

Page 2, line 18: Please, change "underconstrained" to read "unconstrained".

Page 3, line 3: Please, change to read "with respect to".

Page 8, line 16: Please, change to read "parameter sets".

Page 9, line 7: Please, change to read "Pareto fronts".

Page 9, lines 16-17: Please, change to read "The GS metric exhibits a larger variability, thus a larger envelope for both optimizers".

References

Deb, K., Pratap, A., Agarwal, S. Meyarivan, T.: A fast and elitist multiobjective genetic algorithm: NSGA-II. IEEE Trans. Evol. Comput., 6(2), 182-197, 2002.

Efstratiadis, A., and Koutsoyiannis, D.: Fitting hydrological models on multiple responses using the multiobjective evolutionary annealing simplex approach. In: Practical hydroinformatics: Computational intelligence and technological developments in water applications, edited by R.J. Abrahart, L. M. See, and D. P. Solomatine, 259-273, doi:$10.1007/978$-$3$-$540$-$79881$-$1_19, Springer, 2008$.

Efstratiadis, A., and Koutsoyiannis, D.: One decade of multiobjective calibration approaches in hydrological modelling: a review. Hydrological Sciences Journal, 55(1), 58-78, doi:10.1080/02626660903526292, 2010.

Her, Y., and Seong, C.: Responses of hydrological model equifinality, uncertainty, and performance to multi-objective parameter calibration. Journal of Hydroinformatics, 20(4), 864-885, doi:10.2166/hydro.2018.108, 2018.

Sorooshian, S., and Dracup, J. A.: Stochastic parameter estimation procedures for conceptual rainfall-runoff models: Correlated and heteroscedastic error case. Water Resources Research, 16(2), 430-442, doi:10.1029/WR016i002p00430, 1980.

---

## Author Comment (AC1) · 29 Aug 2019

We would like to thank Elena Toth for constructive suggestions on our manuscript that will certainly improve the quality of it. Below we provide the Editor's comments verbatim in black italic text, and our responses are immediately below each comment in blue text.

*Dear Authors, as I anticipated in the submission phase, I have two main suggestions: 1) adding a comparison not only with NSGA-II, but also with the other optimization model you merge in the Caramel (MEAS), since you are proposing an algorithm that should be an improvement above each one of the previous approaches.*

[Figure]

We agree this would have been interesting but we didn't find a code of MEAS available, so we cannot do this comparison for this paper.

*2) the algorithm, in terms of hydrological modelling, is only tested on a single catchment (and in addition, the details of such application are lacking): in order to prove the generality of the improvement allowed by the proposed approach, you should test the calibration of at least another study catchment.*
We will add another example on a new catchment and using the open source hydrological model GR4J (Coron et al., 2017; Coron et al., 2019). We also suggest to change the title of the paper as "Multiobjective calibration by combination of stochastic and gradient-like parameter generation rules: the caRamel algorithm" to be more in the scope of the journal.

*On the abstract/introduction phrasing: p. 1, l. 13: "caRamel()" why the parentheses?*
We will supress this sentence.

*p. 1, l "The comparison with another well-known optimizer (i.e. NSGA-II) confirms the quality of the algorithm"*
We will change it.

*p. 2, ll. 7-14: I would replace with: "CaRamel was initially developed and used for the calibration of hydrological models: Le Moine et al., 2015, Rothfuss et al., 2012, Magand et al., 2014, Monteil et al., 2015 (previously to the R package release) and Rouhier et al. (2017). The interesting performances of caRamel algorithm in such studies prompted us to describe in detail the algorithm in the present paper, and in particular its use as an R package, that can be used for any model in the R environment. The user has simply to define a vector-valued function (at least 2 objectives)*

*for the model to calibrate and lower and upper bounds for the calibrated parameters. This paper aims at describing the principles of caRamel algorithm and its use as an R-package, through the analysis of its results when used for the parametrisation of an hydrological model. A comparison with the widely used NSGA-II (available in the R package, "Multiple Criteria Optimization" MCO, Mersmann et al., 2014), is also presented."*

We will change it.

References

Coron, L., Thirel, G., Delaigue, O., Perrin, C. and Andréassian, V. The Suite of Lumped GR Hydrological Models in an R package. Environmental Modelling and Software, 94, 166-171. DOI: 10.1016/j.envsoft.2017.05.002, 2017.

Coron, L., Delaigue, O., Thirel, G., Perrin, C. and Michel, C. airGR: Suite of GR Hydrological Models for Precipitation-Runoff Modelling. R package version 1.3.2.23. https://CRAN.R-project.org/package=airGR, 2019.

---

## Author Comment (AC2) · 29 Aug 2019

We would like to thank Guillaume Thirel for constructive comments on our manuscript and the script suggestion. The comments are helpful and will certainly improve the quality of our manuscript. Below we provide the Reviewer's comments verbatim in black italic text, and our responses are immediately below each comment in blue text.

*The topic of R packages that are dedicated to hydrology is relevant to HESS, as demonstrated the very recent review paper by Slater et al. (2019) in which I partici- pated. I think that this paper should be mentioned in the manuscript, not because I*

[Figure]

*co-authored it, but because it justifies the interest of such packages to the hydrological community. Indeed, a basic search for the word "package" in the titles or abstracts of HESS papers does not provide so many results.*
Thank you for this very interesting paper, we will add the reference.

*I understand from discussions between the editor and the authors (not available online) that a first version was submitted to another journal and it was suggested to focus more on the package than on actual comparison of algorithms impacts. That can be true for model-oriented journals, such as EM&S and GMD (see for instance our airGR paper in EMS, Coron et al., 2017, which was a Short Communication). However in my opinion HESS is different: describing software and providing pieces of codes is very much appreciate (for the sake of reproducibility), but we also need science. Simply describing packages or tools is not the main aim of HESS. Here the science is a new calibration algorithm and its impact. As a consequence, more emphasis should be put on that two points, but the Editor Elena Toth already commented on that and I agree with her. In the end, I'm wondering if this paper will remain a Technical note or become a full research article.*
We will add a new example of hydrological calibration with GR4J in order to have more results to discuss. We also intend to change the paper title to focus more on the algorithm and less on the R package: "Multiobjective calibration by combination of stochastic and gradient-like parameter generation rules: the caRamel algorithm".

*Overall, the paper is good, it is of interest for the HESS readers in my opinion, and provided that the objectives of the paper are modified as mentioned below and above, I am confident that it will be published later on.*
Thank you.

*Main criticism: My main criticism regards the fact that the authors submitted a*
*manuscript into an open journal, for presenting an open source software, but they illustrate it with an hydrological model that is not open! While I know that MORDOR is the historical EDF hydrological model and using the MORDOR model with caRamel makes sense for the authors, this in my opinion is much less obvious for the readers. I guess that the aim of the authors is to make readers understand that caramel is a valuable tool for calibrating hydrological models available in R. However, no mention is ever made of the fact that there are hydrological models available in the R environment, justifying the interest of this new package. Indeed, readers cannot manipulate the MORDOR model, they only have brief knowledge of the parameters meaning from this paper and they cannot run the lines of codes provided by the authors to handle the package. I think that this is a pity, as the authors made a laudable effort to provide an open source software, some vignettes and also some lines of codes. In addition to the fact that the model is not available, no example data are provided. All of this does not prevent from using the package, but I find it damageable to stop so close to total openness. As one of the developers of the airGR R package, as you know, I tested the caRamel algorithm with a simple GR4J model and an example dataset included in the airGR package. It is not difficult to make it work, but from what was provided it is not straightforward (I guess that the fact that I know well my package helped, but for people using caRamel with a package they did not develop it could be more difficult). I provide the lines of codes I wrote at the end of this review. I would therefore suggest to the authors to include a full example, i.e. one that could be reproduced by the readers, with one open (and well-known if possible!) hydrological model and one open dataset. I don't think this is much work and I don't think that would deeply impact the paper, but I feel that would be useful. Of course, as I provide some lines of codes in this review (see the figures 1, 2, 3, as well as the supplement for the R file) and as some authors know well the GR models, airGR would be a straightforward solution, but as highlighted in Slater et al. (2019) (see section 3.5), other well-known hydrological models are openly available in other R packages.*

Thank you very much for these lines of code. We will add an example of multiobjective

calibration of GR4J, with the data from the R package airGR (Coron et al., 2017; Coron et al., 2019)

*Major remarks:*
*Introduction: nothing is said about composite criteria (e.g. an average of KGEs). I think it should be mentioned and discussed on the base of a couple of references. That's what I tested in the code I provided and the results in terms of performance are quite similar to the multi-objective calibration with the same objectives. This is also much faster, but I guess that the choice of the convergence threshold of caRamel is important in this example.*
The idea of multiobjective calibration is to provide a family of parameter sets and not only one set. We will discuss it with the additional example.

*Heterogeneous spelling in the text: see for instance "Multi Objective" in the title, "multiobjective" and "multiobjective" in the abstract. Or "modelling is the abstract but "modelling" in L. 2, P. 2. "R package" in the title and "R-package" in L. 8, P. 2. "Pareto-front" in P. 5, L. 14 and "Pareto front" in P. 5, L. 28, "Pareto Front" in P. 9, L. 12. Please try to be consistent all over the manuscript! I did not spot all of them here.*
Thank you, we will correct it.

*About section 2.1.1, which describes the generation of parameters. I had difficulties to understand the complete functioning of the 5 rules. First, are these 5 rules or in fact 5 steps that are undertaken successively? In addition, a figure is presented for the first two steps, but it is not used efficiently to make things clear for the readers. I have the feeling that this could be improved easily. For example, why not using only 2 parameters instead of 3, as 3D plots are not possible here? That would simplify things. There are also terms in the legend that are not explained, such as "Example" or "Simplexes". Since the figure is quite complex, with many points, triangles, arrow,*

*it is necessary to help us to read it. I would also maybe suggest a concrete example somehow, especially about the evolution of the parameters sets number, as it is stated at L. 20 that it is necessary to reduce the number of sets. Maybe specify that no parameter set is discarded during the 5 rules stuff?*
Thank you for your suggestion, we will try to clarify this section.

*Section 2.1.2: quite similarly to the previous section, I find that Fig. 2 is underexploited / underexplained. Levels 2 and 3 are never referred to either in the text or in the caption.*
We will add some more explanations.

*Section 4.1: Results are presented for the Kursawe function in Fig. 3. Here again some improvement is possible that would help our reading. What is the optimum for the two objectives? Caption: MCO is not mentioned regarding the number of evaluations; is it 1000? Or does that mean that Fig. 3a only concerns caRamel? If so, why? (I guess this is the case from the reading of the text, but then the caption could be improved). We could have a similar graph for mco.*
Ok, we will correct it.

*Section 4.2.4: Figure 5 is quite difficult to read, in the sense that there are too many points and the superposition of the two algorithms results does not permit to see the differences. Maybe you should separate the two algorithms in two different graphs for each panel? In addition, we don't know from which algorithm are chosen the red points, if I'm not wrong. Red points are chosen from the caRamel optimization, we will mention it in the text. Regarding the name of the package, I recommend, if you use LaTeX, to use the texttt font. This is what we did in Slater et al. (2019) and I think it helps the reading a lot. In addition, it helps making the distinction between the package and the main function, which have the exact same name. We also adopted*

*the spelling "R package" instead of "R-package".*
Thank you for this suggestion but we are not using LaTeX. We will use the spelling "R package".

*Section 3: I would like to have access to the equations of the metrics, as these are not common metrics used by hydrologists.*
We will add the equations.

*References:*
*- Efstratiadis and Koutsoyiannis: this is a conference abstract. What do you think about citing their HSJ paper entitled "One decade of multi-objective calibration approaches in hydrological modelling: A review ?*
We will mention this paper in the introduction and we will add another reference to a book chapter for MEAS.

*- Garavaglia et al. : this paper has been published in 2017, please update.*
We will update it.

*- Le Moine 2009: I think we need a link towards the pdf of this report.*
We will add the link: https://www.metis.upmc.fr/ lemoine/docs/CaRaMEL.pdf.

*- Monteil et al.: any paper or accessible report or presentation instead of this EGU abstract?*
We will add a link toward the presentation.

*- Riquelme et al.: Conférence is written in French* Ok, we will correct it.
*Minor typos and miscellaneous minor stuff:*

*I found a bunch of very minor typos that could easily be dealt with by the authors. As a non-native speaker myself, I hope I'm correct when suggesting these modifications and also that I'm not too picky... I also do not guarantee that I spotted all of them!*

*Page 1:*

*L. 12: remove "algorithm" after NSGA-II as the A of NSGA stands for "algorithm"* ok

*L. 20: The first sentence of the introduction is the same as the first one of the abstract, except for the word "calibration" that is replaced with "prescription". I personally prefer "calibration".* ok

*L. 22: I would add a comma after "precisely"* ok

*L. 23: not 100% sure, but I think that here "statistic" should be spelt "statistics"* ok

*L. 24: "well-knowN"* ok

*Page 2:*

*L. 7-8: please revise the format of citations (i.e. add commas around the years). In addition, I think that the citations should be ordered either by alphabetical or by chronological order.* ok

*L. 8: as I guess this is the case for all citations, I would rephrase the parenthesis as follows: "all previously to the release of the R package"* ok

*L. 9: "of THE caramel algorithm"* ok

*L. 11: I would say "simply has" instead of "has simply"* ok

*L. 17: as it is an adjective to the word "objective", I think that "low-flow" is more correct. At this line and all other occurrences, I would remove the capital letter to "hydrology".* ok

*L. 22: either "one additional objective" or "additional objectiveS". "Constrain", not "constraint".* ok

*L. 24: "THE caRamel algorithm belongs to THE. . ."* ok

*L. 26: "THE caRamel..."* ok

*Page 3:*

*L. 2: I would say "The caRamel algorithm"* ok

*L. 16: "fRont"* ok

*L. 20: "each generation": does each of the 5 rules counts for one generation?*

One generation is created by calling each of the five rules. So one rule is only a part of a generation.

*Page 4:*

*L. 6: "this syntax". In addition, please remove the space after "caRamel" and before the parenthesis.* ok

*L. 9-10 "the minimization or maximization goal": from the way it is written, I was expecting a value. Actually, it is a Boolean used for saying if this is a maximization (minmax = TRUE) or not. I would suggest rephrasing. In addition, I think that the name of the argument, minmax, is not well chosen. minmax is ok if the choices are "min" and "max", but not if the choices are "TRUE" or "FALSE".*

We will put this section as an appendix. We will take your renaming suggestion into account for the next version of the caRamel package.

*L. 15: I would say "choice of a parallel or sequential computation"* ok

*L. 18: not clear. Which kind of managing?*

The user can choose the number of parameters generated by each rule for a generation. For example, this number is set by default to 5 new sets generated by each rules at each generation, but the user may choose to give more weight to some rules.

*L. 19: there are six elements when I tried it, but the last one (gpp) is not explained in the package.*

We will add the description of ggp.

*L. 23: I would say "of objective values", as otherwise we don't know if these are names or values.* ok

*Page 5:*

*L. 6: "an R vignette that gives"* ok

*L. 14, 15, 17: I would add a comma after accuracy, diversity, cardinality.* ok

*L. 20: please add ", which" after "(GS)"* ok

*L. 21: please add a comma after "(HV)" and an hyphen in "volume-based"* ok

*L. 24: I would rather say "the R package* `mco` *("Multiple Criteria Optimization", Mersmann et al., 2014)". Please also make sure in the whole manuscript that the package name is written as mco, not MCO.* ok

*Page 6:*

*L. 2: "with THE Kursawe. . .". In addition, I think it is necessary to introduce a bit what this function, is, what is its aim. I'm not familiar with it and I guess that some other hydrologists also are not.*

We will add a short description of Kursawe function.

*L. 4: "THE population size"* ok

*L. 5 and 8: "THE caRamel ..."* ok

*L. 10: double typo: "show that caRamel"* ok

*Page 7:*

*L. 11: maybe specify on which spatial unit the model is semi-distributed.* In this example, the spatial discretization is based on an elevation zone approach is not used. We will correct it in the text.

*Page 8:*

*L. 2 then is the snow part of the model active?*

No, the snow module is not activated in this example.

*L. 8: define the KGE acronym.* ok

*L. 10: maybe replace "is the result of" with "reflects"* ok

*L. 16: here and everywhere else, I would write "parameter sets" with no S at parameter.* ok

*Page 9:*

*L. 4: "In the mco. . ."* ok

*L. 17: "optimizerS" and "getS"* ok

*L. 18-19: what is the meaning of reproducible? I don't get it.* Reproducibility refers to the ability of one optimizer to give the same results in given conditions. In our case study, the optimization have been run 40 times to check if the results are reproducible.

[Figure]

On Fig. 6c, the envelope with mco is much larger than the one with caRamel, which means that optimization results are more variable with mco.

*Page 10:*

*L. 4: "provide", not "provides"* ok

*L. 5: "give", not "giving"* ok

*L. 6: I quite disagree regarding the "cel" parameter as especially for mco, the spread seems quite large on Fig. 7.*

We were considering the distance between the first and third quartile that is quite narrow.

*L. 13: any idea about how many parameter sets that represents?*

857 for caRamel and 655 for mco.

*Page 11:*

*L. 10: "in an R..."* ok

*L. 11: "THE result"* ok

*L. 14: "from THE mco R package. . ."* ok

*L. 18: "provided that"* ok

*L. 21: "as an R package"* ok

*Appendix A and B:*

*In the code, some spaces are randomly put before or after commas. That could be cleant.* ok

*Table 1: caption: "of THE caRamel. . ."* ok

*I have a couple of comments regarding the descripotion of the main function of the package: - nvar: why not using the word parameter instead of variable? See the description of argument bounds, which uses the name "parameter"* ok

*- minmax: somehow it should appear that several values are possible (e.g. "whether the objectives are. . .")* ok

*- func: "the R objective function"* ok

*- objnames: "nameS"* ok

*- write_gen: that would be much more logical to have a logical here instead of an*

*integer*
We will take that comment into account for the next version of the package.
*- func and funcinit: these two arguments do not take characters but functions. If we enter "kursawe" it does not work, we have to put the name of the function. See FUN argument of tapply for instance.* ok
*Table 2: cetp is a "multiplicative" correction factor* ok

References

Coron, L., Thirel, G., Delaigue, O., Perrin, C. and Andréassian, V. The Suite of Lumped GR Hydrological Models in an R package. Environmental Modelling and Software, 94, 166-171. DOI: 10.1016/j.envsoft.2017.05.002, 2017.

Coron, L., Delaigue, O., Thirel, G., Perrin, C. and Michel, C. airGR: Suite of GR Hydrological Models for Precipitation-Runoff Modelling. R package version 1.3.2.23. https://CRAN.R-project.org/package=airGR, 2019.

---

## Author Comment (AC3) · 29 Aug 2019

We would like to thank Andreas Efstratiadis for constructive comments on our manuscript. The comments are helpful and will certainly improve the quality of the paper. Below we provide the Reviewer's comments verbatim in black italic text, and our responses are immediately below each comment in blue text.

*My overall opinion about this article is positive, yet in its current form cannot stand neither as a technical note not as research paper. Actually, it rather resembles to an extended abstract of a clearly hard and long research, which may be useful as a brief*

[Figure]

*documentation for the R community, but is not suitable for a top hydrological journal such as HESS. First of all, the authors have to decide the orientation and the objectives of this article. There are several alternatives, i.e. (a) a state-of-the-art discussion of the multiobjective calibration problem in hydrology; (b) a comprehensive description and justification of the algorithm and its technological advances, accompanied by extended tests of its performance against problems of varying complexity and against other well established methodologies, and (c) a more synoptic description of the algorithm, with emphasis to its application to few (not only one) representative hydrological calibration problems of varying difficulty, to be presented and discussed in detail.*

We intend to clarify the orientation of the article with the alternative (c), by giving more details on the algorithm and by adding a new application in hydrology with the code GR4J (Coron et al., 2017; Coron et al., 2019). We also suggest to change the title as "Multiobjective calibration by combination of stochastic and gradient-like parameter generation rules: the caRamel algorithm".

*Specific comments*
*Page 1, lines 13-14: "The main function of the package, caRamel(), requires to define a multi-objective calibration function as well as bounds on the variation of the underlying parameters to optimize". Too obvious technical detail to be referred in the abstract.*

Thank you, we will correct it.

*Page 1, lines 24-25: ". . . it is well-know that errors in a simulated discharge time series are not normally distributed, and do not have constant variance and autocorrelation." This statement is true (a reference would be helpful, e.g. Sorooshian and Dracup, 1980), but is not so much evidently linked with the need for multiobjective calibration. Actually, the multiobjective approach in hydrological modelling covers much more cases, including fitting to multivariable and multisite data, as well as soft information*

*(cf., Madsen, 2003; Efstratiadis Koutsoyiannis, 2010).*
We will add the references you mentioned.

*Page 1, lines 28-29: "Evolutionary algorithms have become widely used to explore the Pareto-optimal front in multi-objective optimization problems that are too complex to be solved by descent methods". Do they exist descent methods for multiobjective optimization? Maybe you refer to classical aggregation approaches (e.g. weighting of criteria) that have to be solved multiple times with different weighting values, in contrast to evolutionary approaches that only require "a single optimization run", as correctly mentioned just after (page 2, line 2).*
Yes, we refer to descent methods with aggregation approaches.

*Page 2, lines 5-6: "The caRamel optimizer has been developed to meet the need of an automatic calibration procedure that delivers not only one but a family of parameters sets that are optimal regarding a multi-objective target". There do exist many algorithms covering this general objective. Is there any specific objective for the development of caRamel? Which shortcomings of the existing algorithms have you detected before deciding building a new method?*
Our feeling is that most of MOEA rely mainly on stochastic generation rules, with few deterministic aspects. The idea in caRamel is of course to keep these stochastic, "global" mechanisms (such as recombination or multivariate sampling using the covariance) but also to make place for more "local" mechanisms, such as extrapolation along vectors in the parameter space which are associated with an improvement in all objective functions (a "gradient-like" approach extended to the set of objective functions, in a qualitative way). A shared feature between caRamel and MEAS is the use of the simplexes in which generational rules are applied. However, in MEAS these simplexes are randomly chosen, with the sole constraint that at least one vertex is in the approximated Pareto front; conversely, in caRamel the choice of the simplexes

is entirely deterministic since they are the result of the Delaunay triangulation of the individuals in the objective space (with each objective scaled by the specified precision), and the probability of using a given simplex for generating new individuals is proportional to the volume of this simplex in the scaled objective space. The same kind of geometrical rationale applies for the selection of edges along which an "expansion" is tested (see the description of the rules when added).

*Page 2, line 17: Terms "flood objective" and "low flow objective" are unclear (at least for a non-expert).*
We will give more detail.

*Page 2, lines 17-19: "Multi-objective calibration is also a way to add some constraints to an underconstrained problem when many parameters have to be quantified. This can help to reduce the equifinality of parameters sets". More discussion should be made here (for 30 years, equifinality remains a hot topic in hydrology), including some representative references, e.g. Her and Seong (2018).*
We will add the reference.

*Page 2, lines 20-21: "Equifinality may be caused by the model structure, when two sets of parameters give similar results. Another kind of equifinality is related to the calibration objectives, when two different model results give similar objective values." Term "result" is unclear - probably you refer to the model outputs, by means of response time series. In this respect, two different parameter sets, except if they are very close, cannot provide the same outputs (i.e., similar individual values), they can provide outputs with similar statistical characteristics, and thus similar performance metrics, as correctly stated in the second phrase.*
We propose to rephrase: "Equifinality may be caused by the model structure, when two sets of parameters give similar model outputs due to interactions between model

parameters."

*Page 2, line 28: Please, also cite the more detailed and peer-reviewed paper by Efstratiadis and Koutsoyiannis (2008), published as book chapter.*
We will add the reference.

*Page 3, section 2.1.1 (Generation rules): The description of the algorithm is very poor and only provides a very general idea about the generation mechanisms. How are these rules associated with the ones used in MEAS? I see quite many differences and very interesting ideas implemented here, but the text is too short to allow understanding and evaluating the methodology (and its potential novelties). Figure 1 is also little helpful; for instance, green and blue points, indicating new sets, are missing, although they are referred in the legend.*
We will expand this section and try to clarify it.

*Page 3, line 25: Why you keep points of the lower level? Aren't they dominated by points of the upper one?*
The non-dominated level number is 1, so points of upper level are dominated by points of lower levels.

*Pages 4-5, section 2.2 (The caRamel R package): This section is very technical and not so much relevant with the broader philosophy of HESS.*
We propose to move it to an appendix.

*Page 5, line 15: "The diversity which can be described with two aspects: the spread of the set . . ." Diversity may refer both to the parameter and the objective space. Which of the two sets are used here?*

We refer to diversity in the objective space.

*Page 5, line 25: Please, cite Deb et al. (2002) who developed NSGA-II.*
We will add the reference.

*Page 6, lines 12-13: "Comparison with MCO (NSGA-II only) shows that the use of MEAS makes the optimization process converge more rapidly but with a lower diversity". Can you explain the reason of this behavior? Is this an inherent drawback of caRamel, or is due to the algorithmic inputs used in this experiment? As shown in Table 1 (and similarly to all hybrid optimization schemes), caRamel uses quite a large number of input arguments that need manual tuning. Did you run the algorithm by testing alternative set-ups? Do you have any recommendations for the users, regarding the selection of these inputs?*
The user may choose to give more weight to some of the rules in the input arguments. We tested different combination but our feeling is that it is better to have a "balanced" approach with the same number of parameter sets generated for each rule.

*Page 7, section 4.2 (Hydrological modeling): Your case study does not allow extracting safe conclusions about the performance of your method and its comparison against NSGA II. The key reason is that the use of a single overall metric, i.e. KGE, ensures almost perfect fitting to observations (KGE = 95).*
The KGE had been computed on 3 different flow signatures, so we consider them as 3 metrics. For the other example we plan to add, we will also use Nash-Sutcliffe Efficiency as a metric.

*Page 8, line 8: Please, better explain criteria (2) and (3) and the associated signatures. Have been these criteria used elsewhere? If yes, please also provide the associated*

*references.*
These criteria are also used in Rouhier et al (2017). We will add the reference.

*Page 9, lines 3-4: "MCO has been used with crossover probability set to 0.5 and mutation probability to 0.3". Have you made any preliminary tests before selective these values? Which are the values applied to the input arguments of caRamel?*
Please find below (Fig. 1) a heat map of the Hypervolume metric regarding cprob and mprob for the hydrological model calibration with mco, that helps us to chose the values. We will add the values of the input arguments for caRamel.

*Page 9, Figure 5: I find your figure a little bit misleading. In the vertical axis, the spread of solutions is very small, and within the anticipated range of uncertainty induced in any hydrological calibration exercise. For instance, the lower value of KGEamd is 0.83, while the higher is 0.86. From my point-of-view, such differences do not make sense in the real world.*
For the additional hydrological example, we will not use the KGEamd objective.

*Page 10, line 12: How did you selected the best compromise parameter set? What do you mean by term "observed set"?*
The best-compromise set has been selected regarding to the distance to the point (1,1,1) in the objective space. The "observed one" stand for measured streamflow. We will rephrase it.

*Page 11, section 5 (Conclusions): This section is poorly developed. It has to be written from scratch, to highlight the advantages and weaknesses of the methodology and also discuss ideas for future research.*
We will rewrite it, and also add conclusions from the additional example.

*Minor editorial comments*
*Page 2, line 10: In which of the aforementioned papers do you describe the algorithm?*
*It is not clear here.*
These papers refer to research work using caRamel but the algorithm itself was not described.

*Page 2, line 17: Term "Hydrology" should not start with capital.*
We will correct it.

*Page 2, line 18: Please, change "underconstrained" to read "unconstrained".*
We will change it.

*Page 3, line 3: Please, change to read "with respect to".*
We will change it.

*Page 8, line 16: Please, change to read "parameter sets".*
We will change it.

*Page 9, line 7: Please, change to read "Pareto fronts".*
We will change it.

*Page 9, lines 16-17: Please, change to read "The GS metric exhibits a larger variability, thus a larger envelope for both optimizers".*
We will change it.

[Figure]

**References**

Coron, L., Thirel, G., Delaigue, O., Perrin, C. and Andréassian, V. The Suite of Lumped GR Hydrological Models in an R package. Environmental Modelling and Software, 94, 166-171. DOI: 10.1016/j.envsoft.2017.05.002, 2017.

Coron, L., Delaigue, O., Thirel, G., Perrin, C. and Michel, C. airGR: Suite of GR Hydrological Models for Precipitation-Runoff Modelling. R package version 1.3.2.23. https://CRAN.R-project.org/package=airGR, 2019.

Rouhier, L., Le Lay, M., Garavaglia, F., Le Moine, N., Hendrickx, F., Monteil, C., and Ribstein, P.: Impact of mesoscale spatial variability of climatic inputs and parameters on the hydrological response. Journal of Hydrology 553, 13-25. http://dx.doi.org/10.1016/j.jhydrol.2017.07.037, 2017.

Heat map with axes: y-axis labeled "mprob" ranging 0.00 to 1.00, x-axis labeled "cprob" ranging 0.00 to 1.00. Legend labeled "value" with scale 0.0111, 0.0109, 0.0107, 0.0105.

**Fig. 1.** Heat map of Hypervolume for mco regarding cprob and mprob (calibration of Mordor-SD, catchment Tarn at Millau).

---

## Author Response (AR1)

We would like to thank the editor Elena Toth and the two referees Guillaume Thirel and Andreas Efstratiadis for their constructive suggestions and comments. Below we provide the Editor's and Referee's comments verbatim in black italic text and our responses below each comment in blue text.

**1   Response to editor decision**

*Dear Authors, I would like to warmly thank the two Referees, who have uploaded their detailed and very constructive comments very timely and are both waiting for your revised version, to be written accordingly to the intentions you have listed in your replies. Their comments were, as I expected, mainly positive on the content/value of presenting the proposed algorithm, but asking a major revision of the applications and of the presentation.*

We would also thank the two Referees who have given very helpful recommendations. The major changes in our paper are the following points :

- we want to change the title of our paper to "Multi-objective calibration by combination of stochastic and gradient-like parameter generation rules: the caRamel algorithm" in order to focus on the algorithm itself,

- we added a more detailed description of the algorithm,

- we added a comparison with MEAS for each case study, as Andreas Efstratiadis kindly gave us a code for MEAS,

- we added an example of calibration of an open source hydrological model with open source data (GR4J model, as suggested by Guillaume Thirel).

The paper has been proofread by a native English speaker. The marked-up manuscript version is given bellow.

*I keep thinking that you need a comparison not only with NSGA but also with MEAS, which is more similar in its concepts than the NSGA: perhaps our Referee Andreas Efstratiadis may help you in this.*

This comparison has been added for all the selected case studies.

*And I look forward to see the application of both the open-source hydrological model and MORDOR over more than one catchment: since you chose option c) among the alternatives given by A. Efstratiadis, the paper needs an interpretation of the results you will obtain both changing the model and changing the catchment.*

We add the calibration of open source hydrological model GR4J on a fictional catchment (Coron et al., 2017, 2019) and we also present the calibration of MORDOR-TS model on two contrasted catchments (one mainly pluvial, the other with snow influence).

*I agree with the referees on the need to improve/explain better: - The 5 rules (section 2.1.1)*

This is done in the new section 3.1

*- The relationship between Caramel and the two 'inspiring' algorithms*

The section 3 has been expanded.

*- The metrics for assessing optimization algorithm (section 3)*

This is done on section 4.2

*- The meaning of the algorithm internal parameters/weights and the reasons for their choice (inside the Kursawe test function section, where their impact may be examined). Show impact of using different values for such parameters (not only 'feelings')*

We have run many optimizations to select the more appropriate configuration for each optimizer. We then chose to present only one configuration by case study to have more readable results.

45     *- The hydrological models (once you will have two): while GR is well-known and plenty of documentation is accessible, whereas for MORDOR you need to better explain how it works and especially its differences from GR.*
    The main difference between GR4J and MORDOR-TS is that the model is spacially distributed. MORDOR-TS model is presented in Rouhier et al. (2017).

50     *- The catchments (once you will have two): their climate, dominant hydrological processes and differences (possibly chose catchment with different hydro-climatic regimes)*
    We chose two French catchments Tarn River at Millau and Durance River at La Clapière. The Tarn River regime is pluvial while Durance River regime is strongly influenced by the snow. We also add a graph of hydrological regime for each catchment (Fig. 6).

55

    *On A. Efstratiadis' comment on p. 1, ll 24-25: it is not a matter of adding the references but of explaining the main reasons for applying a multi-objective calibration, that is not solely due to addressing the statistical properties of the errors.*
    We added this sentence: "In addition, Efstratiadis and Koutsoyiannis (2010) list other advantages of multi-objective calibration such as ensure parsimony between the number of objectives against parameters to optimize, fit distributed responses
60 of models on multiple measurements, recognize the uncertainties and structural errors related to model configuration and the parameters estimation procedure, handling objectives having contradictory performance."'

    *On hydrological modelling performances (A. Efstratiadis' comment on p. 7): I would suggest, rather than adding Nash-Sutcliffe, to analyse the three separate components of the KGE.*
65     We followed this suggestion for the calibration of GR4J model.

**2   Response to interactive comment of Editor Elena Toth**

*Dear Authors, as I anticipated in the submission phase, I have two main suggestions: 1) adding a comparison not only with NSGA-II, but also with the other optimization model you merge in the Caramel (MEAS), since you are proposing an algorithm that should be an improvement above each one of the previous approaches.*
70     We added the comparison with MEAS.

    *2) the algorithm, in terms of hydrological modelling, is only tested on a single catchment (and in addition, the details of such application are lacking): in order to prove the generality of the improvement allowed by the proposed approach, you should test the calibration of at least another study catchment.*
75     We added another example on a new catchment and using the open source hydrological model GR4J (Coron et al., 2017; Coron et al., 2019). We also suggest to change the title of the paper as "Multiobjective calibration by combination of stochastic and gradient-like parameter generation rules: the caRamel algorithm" to be more in the scope of the journal.

    *On the abstract/introduction phrasing: p. 1, l. 13: "caRamel()" why the parentheses?*
80     We suppressed this sentence.

    *p. 1, l "The comparison with another well-known optimizer (i.e. NSGA-II) confirms the quality of the algorithm"*
    We wrote: "The comparison with other optimizers on hydrological case studies (i.e. NSGA-II, MEAS) confirms the quality of the algorithm."

85

    *p. 2, ll. 7-14: I would replace with: "CaRamel was initially developed and used for the calibration of hydrological models: Le Moine et al., 2015, Rothfuss et al., 2012, Magand et al., 2014, Monteil et al., 2015 (previously to the R package release) and Rouhier et al. (2017). The interesting performances of caRamel algorithm in such studies prompted us to describe in detail the algorithm in the present paper, and in particular its use as an R package, that can be used for any model in the R environment.*
90 *The user has simply to define a vector-valued function (at least 2 objectives) for the model to calibrate and lower and upper*

*bounds for the calibrated parameters. This paper aims at describing the principles of caRamel algorithm and its use as an R-package, through the analysis of its results when used for the parametrisation of an hydrological model. A comparison with the widely used NSGA-II (available in the R package, "Multiple Criteria Optimization" MCO, Mersmann et al., 2014), is also presented."*

95     We corrected id

**3    Response to interactive comment of Referee Guillaume Thirel**

*The topic of R packages that are dedicated to hydrology is relevant to HESS, as demonstrated the very recent review paper by Slater et al. (2019) in which I participated. I think that this paper should be mentioned in the manuscript, not because I*
100 *co-authored it, but because it justifies the interest of such packages to the hydrological community. Indeed, a basic search for the word "package" in the titles or abstracts of HESS papers does not provide so many results.*
    Thank you for this very interesting paper, we added the reference.

*I understand from discussions between the editor and the authors (not available online) that a first version was submitted to*
105 *another journal and it was suggested to focus more on the package than on actual comparison of algorithms impacts. That can be true for model-oriented journals, such as EM&S and GMD (see for instance our airGR paper in EMS, Coron et al., 2017, which was a Short Communication). However in my opinion HESS is different: describing software and providing pieces of codes is very much appreciate (for the sake of reproducibility), but we also need science. Simply describing packages or tools is not the main aim of HESS. Here the science is a new calibration algorithm and its impact. As a consequence, more emphasis*
110 *should be put on that two points, but the Editor Elena Toth already commented on that and I agree with her. In the end, I'm wondering if this paper will remain a Technical note or become a full research article.*
    We added a new example of hydrological calibration with GR4J in order to have more results to discuss. We also changed the paper title to focus more on the algorithm and less on the R package: "Multiobjective calibration by combination of stochastic and gradient-like parameter generation rules: the caRamel algorithm".
115

*Overall, the paper is good, it is of interest for the HESS readers in my opinion, and provided that the objectives of the paper are modified as mentioned below and above, I am confident that it will be published later on.*
    Thank you.

120 *Main criticism: My main criticism regards the fact that the authors submitted a manuscript into an open journal, for presenting an open source software, but they illustrate it with an hydrological model that is not open! While I know that MORDOR is the historical EDF hydrological model and using the MORDOR model with caRamel makes sense for the authors, this in my opinion is much less obvious for the readers. I guess that the aim of the authors is to make readers understand that caramel is a valuable tool for calibrating hydrological models available in R. However, no mention is ever made of the fact that there*
125 *are hydrological models available in the R environment, justifying the interest of this new package. Indeed, readers cannot manipulate the MORDOR model, they only have brief knowledge of the parameters meaning from this paper and they cannot run the lines of codes provided by the authors to handle the package. I think that this is a pity, as the authors made a laudable effort to provide an open source software, some vignettes and also some lines of codes. In addition to the fact that the model is not available, no example data are provided. All of this does not prevent from using the package, but I find it damageable to*
130 *stop so close to total openness. As one of the developers of the airGR R package, as you know, I tested the caRamel algorithm with a simple GR4J model and an example dataset included in the airGR package. It is not difficult to make it work, but from what was provided it is not straightforward (I guess that the fact that I know well my package helped, but for people using caRamel with a package they did not develop it could be more difficult). I provide the lines of codes I wrote at the end of this review. I would therefore suggest to the authors to include a full example, i.e. one that could be reproduced by the readers, with*
135 *one open (and well-known if possible!) hydrological model and one open dataset. I don't think this is much work and I don't think that would deeply impact the paper, but I feel that would be useful. Of course, as I provide some lines of codes in this*

*review (see the figures 1, 2, 3, as well as the supplement for the R file) and as some authors know well the GR models, airGR would be a straightforward solution, but as highlighted in Slater et al. (2019) (see section 3.5), other well-known hydrological models are openly available in other R packages.*

Thank you very much for these lines of code. We added an example of multiobjective calibration of GR4J, with the data from the R package `airGR` (Coron et al., 2017; Coron et al., 2019)

*Major remarks:*
*Introduction: nothing is said about composite criteria (e.g. an average of KGEs). I think it should be mentioned and discussed on the base of a couple of references. That's what I tested in the code I provided and the results in terms of performance are quite similar to the multi-objective calibration with the same objectives. This is also much faster, but I guess that the choice of the convergence threshold of caRamel is important in this example.*

The idea of multiobjective calibration is to provide a family of parameter sets and not only one set. This results in a envelope of simulated discharges whose parameters sets are on the Pareto Front (Fig. 11).

*Heterogeneous spelling in the text: see for instance "Multi Objective" in the title, "multiobjective" and "multiobjective" in the abstract. Or "modelling is the abstract but "modelling" in L. 2, P. 2. "R package" in the title and "R-package" in L. 8, P. 2. "Pareto-front" in P. 5, L. 14 and "Pareto front" in P. 5, L. 28, "Pareto Front" in P. 9, L. 12. Please try to be consistent all over the manuscript! I did not spot all of them here.*

Thank you, we corrected it.

*About section 2.1.1, which describes the generation of parameters. I had difficulties to understand the complete functioning of the 5 rules. First, are these 5 rules or in fact 5 steps that are undertaken successively? In addition, a figure is presented for the first two steps, but it is not used efficiently to make things clear for the readers. I have the feeling that this could be improved easily. For example, why not using only 2 parameters instead of 3, as 3D plots are not possible here? That would simplify things. There are also terms in the legend that are not explained, such as "Example" or "Simplexes". Since the figure is quite complex, with many points, triangles, arrow, it is necessary to help us to read it. I would also maybe suggest a concrete example somehow, especially about the evolution of the parameters sets number, as it is stated at L. 20 that it is necessary to reduce the number of sets. Maybe specify that no parameter set is discarded during the 5 rules stuff?*

We expanded this section to give more explanations.

*Section 2.1.2: quite similarly to the previous section, I find that Fig. 2 is underexploited / underexplained. Levels 2 and 3 are never referred to either in the text or in the caption.*

Level 2 and 3 are dominated levels.

*Section 4.1: Results are presented for the Kursawe function in Fig. 3. Here again some improvement is possible that would help our reading. What is the optimum for the two objectives? Caption: MCO is not mentioned regarding the number of evaluations; is it 1000? Or does that mean that Fig. 3a only concerns caRamel? If so, why? (I guess this is the case from the reading of the text, but then the caption could be improved). We could have a similar graph for mco.*

We add Figure 7 with Pareto front from all optimizers

*Section 4.2.4: Figure 5 is quite difficult to read, in the sense that there are too many points and the superposition of the two algorithms results does not permit to see the differences. Maybe you should separate the two algorithms in two different graphs for each panel? In addition, we don't know from which algorithm are chosen the red points, if I'm not wrong. Red points are chosen from the caRamel optimization, we will mention it in the text. Regarding the name of the package, I recommend, if you use LaTeX, to use the texttt font. This is what we did in Slater et al. (2019) and I think it helps the reading a lot. In addition, it helps making the distinction between the package and the main function, which have the exact same name. We also adopted the spelling "R package" instead of "R-package".*

Thank you for this suggestion, we use this package for the new version of the paper.

*Minor typos and miscellaneous minor stuff:*
*I found a bunch of very minor typos that could easily be dealt with by the authors. As a non-native speaker myself, I hope I'm correct when suggesting these modifications and also that I'm not too picky... I also do not guarantee that I spotted all of them!*
*Page 1:*
*L. 12: remove "algorithm" after NSGA-II as the A of NSGA stands for "algorithm"* ok
*L. 20: The first sentence of the introduction is the same as the first one of the abstract, except for the word "calibration" that is replaced with "prescription". I personally prefer "calibration".* ok
*L. 22: I would add a comma after "precisely"* ok
*L. 23: not 100% sure, but I think that here "statistic" should be spelt "statistics"* ok
*L. 24: "well-knowN"* ok
*Page 2:*
*L. 7-8: please revise the format of citations (i.e. add commas around the years). In addition, I think that the citations should be ordered either by alphabetical or by chronological order.* ok
*L. 8: as I guess this is the case for all citations, I would rephrase the parenthesis as follows: "all previously to the release of the R package"* ok
*L. 9: "of THE caramel algorithm"* ok
*L. 11: I would say "simply has" instead of "has simply"* ok
*L. 17: as it is an adjective to the word "objective", I think that "low-flow" is more correct. At this line and all other occurrences, I would remove the capital letter to "hydrology".* ok
*L. 22: either "one additional objective" or "additional objectiveS". "Constrain", not "constraint".* ok
*L. 24: "THE caRamel algorithm belongs to THE. . ."* ok
*L. 26: "THE caRamel..."* ok
*Page 3:*
*L. 2: I would say "The caRamel algorithm"* ok
*L. 16: "fRont"* ok
*L. 20: "each generation": does each of the 5 rules counts for one generation?*
One generation is created by calling each of the five rules. So one rule is only a part of a generation.
*Page 4:*
*L. 6: "this syntax". In addition, please remove the space after "caRamel" and before the parenthesis.* ok
*L. 9-10 "the minimization or maximization goal": from the way it is written, I was expecting a value. Actually, it is a Boolean used for saying if this is a maximization (minmax = TRUE) or not. I would suggest rephrasing. In addition, I think that the name of the argument, minmax, is not well chosen. minmax is ok if the choices are "min" and "max", but not if the choices are "TRUE" or "FALSE".*
We put this section as an appendix. We will take your renaming suggestion into account for the next version of the caRamel

235     package.

*L. 15: I would say "choice of a parallel or sequential computation"* ok

*L. 18: not clear. Which kind of managing?*

The user can choose the number of parameters generated by each rule for a generation. For example, this number is set by default to 5 new sets generated by each rules at each generation, but the user may choose to give more weight to some rules.

240   *L. 19: there are six elements when I tried it, but the last one (gpp) is not explained in the package.*

We added the description of ggp.

*L. 23: I would say "of objective values", as otherwise we don't know if these are names or values.* ok

*Page 5:*

*L. 6: "an R vignette that gives"* ok

245   *L. 14, 15, 17: I would add a comma after accuracy, diversity, cardinality.* ok

*L. 20: please add ", which" after "(GS)"* ok

*L. 21: please add a comma after "(HV)" and an hyphen in "volume-based"* ok

*L. 24: I would rather say "the R package* mco *("Multiple Criteria Optimization", Mersmann et al., 2014)". Please also make sure in the whole manuscript that the package name is written as mco, not MCO.* ok

250   *Page 6:*

*L. 2: "with THE Kursawe...". In addition, I think it is necessary to introduce a bit what this function, is, what is its aim. I'm not familiar with it and I guess that some other hydrologists also are not.*

We added a short description of Kursawe function in section 4.3.1.

*L. 4: "THE population size"* ok

255   *L. 5 and 8: "THE caRamel ..."* ok

*L. 10: double typo: "show that caRamel"* ok

*Page 7:*

*L. 11: maybe specify on which spatial unit the model is semi-distributed.* In this example, the spatial discretization is based on an elevation zone approach is not used. We will correct it in the text.

260   *Page 8:*

*L. 2 then is the snow part of the model active?*

A simplified snow model is active for the Tarn at Millau case study.

*L. 8: define the KGE acronym.* ok

*L. 10: maybe replace "is the result of" with "reflects"* ok

265   *L. 16: here and everywhere else, I would write "parameter sets" with no S at parameter.* ok

*Page 9:*

*L. 4: "In the mco. . ."* ok

*L. 17: "optimizerS" and "getS"* ok

*L. 18-19: what is the meaning of reproducible? I don't get it.* Reproducibility refers to the ability of one optimizer to give

270   the same results in given conditions. In our case study, the optimization have been run 40 times to check if the results are reproducible. On Fig. 6c, the envelope with mco is much larger than the one with caRamel, which means that optimization results are more variable with mco.

*Page 10:*

*L. 4: "provide", not "provides"* ok

275   *L. 5: "give", not "giving"* ok

*L. 6: I quite disagree regarding the "cel" parameter as especially for mco, the spread seems quite large on Fig. 7.*

We were considering the distance between the first and third quartile that is quite narrow.

*L. 13: any idea about how many parameter sets that represents?*

We added the number of Pareto vectors for each optimizer on the beginning of section 5.1.

280   *Page 11:*

*L. 10: "in an R..."* ok

*L. 11: "THE result"* ok

*L. 14: "from THE mco R package. . ."* ok

*L. 18: "provided that"* ok

285 *L. 21: "as an R package"* ok

*Appendix A and B:*

*In the code, some spaces are randomly put before or after commas. That could be cleant.* ok

*Table 1: caption: "of THE caRamel. . ."* ok

*I have a couple of comments regarding the description of the main function of the package: - nvar: why not using the word*

290 *parameter instead of variable? See the description of argument bounds, which uses the name "parameter"* ok

*- minmax: somehow it should appear that several values are possible (e.g. "whether the objectives are. . .")* ok

*- func: "the R objective function"* ok

*- objnames: "nameS"* ok

*- write_gen: that would be much more logical to have a logical here instead of an integer*

295 We will take that comment into account for the next version of the package.

*- func and funcinit: these two arguments do not take characters but functions. If we enter "kursawe" it does not work, we have to put the name of the function. See FUN argument of tapply for instance.* ok

*Table 2: cetp is a "multiplicative" correction factor* ok

300 ## 4    Response to interactive comment of Referee Andreas Efstratiadis

*My overall opinion about this article is positive, yet in its current form cannot stand neither as a technical note not as research paper. Actually, it rather resembles to an extended abstract of a clearly hard and long research, which may be useful as a brief documentation for the R community, but is not suitable for a top hydrological journal such as HESS. First of all, the authors have to decide the orientation and the objectives of this article. There are several alternatives, i.e. (a) a state-of-the-*

305 *art discussion of the multiobjective calibration problem in hydrology; (b) a comprehensive description and justification of the algorithm and its technological advances, accompanied by extended tests of its performance against problems of varying complexity and against other well established methodologies, and (c) a more synoptic description of the algorithm, with emphasis to its application to few (not only one) representative hydrological calibration problems of varying difficulty, to be presented and discussed in detail.*

310 We orientated this new version of the paper with the alternative (c), by giving more details on the algorithm and by adding a two new applications in hydrology with the code GR4J (Coron et al., 2017; Coron et al., 2019) and the calibration of a snowy catchment. We also want to change the title as "Multiobjective calibration by combination of stochastic and gradient-like parameter generation rules: the caRamel algorithm".

315 *Specific comments*

*Page 1, lines 13-14: "The main function of the package, caRamel(), requires to define a multi-objective calibration function as well as bounds on the variation of the underlying parameters to optimize". Too obvious technical detail to be referred in the abstract.*

Thank you, we removed it.

320

*Page 1, lines 24-25: ". . . it is well-know that errors in a simulated discharge time series are not normally distributed, and do not have constant variance and autocorrelation." This statement is true (a reference would be helpful, e.g. Sorooshian and Dracup, 1980), but is not so much evidently linked with the need for multiobjective calibration. Actually, the multiobjective approach in hydrological modelling covers much more cases, including fitting to multivariable and multisite data, as well as*

325 *soft information (cf., Madsen, 2003; Efstratiadis Koutsoyiannis, 2010).*

We will added the references you mentioned and other application of multi-objective calibration.

*Page 1, lines 28-29: "Evolutionary algorithms have become widely used to explore the Pareto-optimal front in multi-objective optimization problems that are too complex to be solved by descent methods". Do they exist descent methods for multiobjective*

330 *optimization? Maybe you refer to classical aggregation approaches (e.g. weighting of criteria) that have to be solved multiple times with different weighting values, in contrast to evolutionary approaches that only require "a single optimization run", as correctly mentioned just after (page 2, line 2).*

Yes, we refer to descent methods with aggregation approaches.

335 *Page 2, lines 5-6: "The caRamel optimizer has been developed to meet the need of an automatic calibration procedure that delivers not only one but a family of parameters sets that are optimal regarding a multi-objective target". There do exist many algorithms covering this general objective. Is there any specific objective for the development of caRamel? Which shortcomings of the existing algorithms have you detected before deciding building a new method?*

Our feeling is that most of MOEA rely mainly on stochastic generation rules, with few deterministic aspects. The idea in
340 caRamel is of course to keep these stochastic, "global" mechanisms (such as recombination or multivariate sampling using the covariance) but also to make place for more "local" mechanisms, such as extrapolation along vectors in the parameter space which are associated with an improvement in all objective functions (a "gradient-like" approach extended to the set of objective functions, in a qualitative way). A shared feature between caRamel and MEAS is the use of the simplexes in which generational rules are applied. However, in MEAS these simplexes are randomly chosen, with the sole constraint that at least one vertex is
345 in the approximated Pareto front; conversely, in caRamel the choice of the simplexes is entirely deterministic since they are the result of the Delaunay triangulation of the individuals in the objective space (with each objective scaled by the specified precision), and the probability of using a given simplex for generating new individuals is proportional to the volume of this simplex in the scaled objective space. The same kind of geometrical rationale applies for the selection of edges along which an "expansion" is tested (see the description of the rules section 3.1).

350

*Page 2, line 17: Terms "flood objective" and "low flow objective" are unclear (at least for a non-expert).*
We changed the objectives of calibration to Nash-Sutcliffe and Kling-Gupta Efficiencies at two internal gauging points and at the oulet .

355 *Page 2, lines 17-19: "Multi-objective calibration is also a way to add some constraints to an underconstrained problem when many parameters have to be quantified. This can help to reduce the equifinality of parameters sets". More discussion should be made here (for 30 years, equifinality remains a hot topic in hydrology), including some representative references, e.g. Her and Seong (2018).*

We added: "Her and Seong (2018) showed that the introduction of an adequate number of objective functions could improve
360 the quality of calibration without requiring additional observations. The amount of equifinality and output uncertainty overall decreased while the model performance was maintained as the number of objective functions increased sequentially until four objective functions."

*Page 2, lines 20-21: "Equifinality may be caused by the model structure, when two sets of parameters give similar results.*
365 *Another kind of equifinality is related to the calibration objectives, when two different model results give similar objective values." Term "result" is unclear - probably you refer to the model outputs, by means of response time series. In this respect, two different parameter sets, except if they are very close, cannot provide the same outputs (i.e., similar individual values), they can provide outputs with similar statistical characteristics, and thus similar performance metrics, as correctly stated in the second phrase.*
370 We propose to rephrase: "Equifinality may be caused by the model structure, when two sets of parameters give similar model outputs due to interactions between model parameters."

*Page 2, line 28: Please, also cite the more detailed and peer-reviewed paper by Efstratiadis and Koutsoyiannis (2008), published as book chapter.*
375 We added the reference.

*Page 3, section 2.1.1 (Generation rules): The description of the algorithm is very poor and only provides a very general idea about the generation mechanisms. How are these rules associated with the ones used in MEAS? I see quite many differences*

*and very interesting ideas implemented here, but the text is too short to allow understanding and evaluating the methodology*
*(and its potential novelties). Figure 1 is also little helpful; for instance, green and blue points, indicating new sets, are missing,*
*although they are referred in the legend.*
        We added the section 3.1 for the rules description.

*Page 3, line 25: Why you keep points of the lower level? Aren't they dominated by points of the upper one?*
        The non-dominated level number is 1, so points of upper level are dominated by points of lower levels.

*Pages 4-5, section 2.2 (The caRamel R package): This section is very technical and not so much relevant with the broader*
*philosophy of HESS.*
        We moved it to appendix A.

*Page 5, line 15: "The diversity which can be described with two aspects: the spread of the set . . ." Diversity may refer both*
*to the parameter and the objective space. Which of the two sets are used here?*
        We refer to diversity in the objective space.

*Page 5, line 25: Please, cite Deb et al. (2002) who developed NSGA-II.*
        We added the reference.

*Page 6, lines 12-13: "Comparison with MCO (NSGA-II only) shows that the use of MEAS makes the optimization process*
*converge more rapidly but with a lower diversity". Can you explain the reason of this behavior? Is this an inherent drawback*
*of caRamel, or is due to the algorithmic inputs used in this experiment? As shown in Table 1 (and similarly to all hybrid*
*optimization schemes), caRamel uses quite a large number of input arguments that need manual tuning. Did you run the*
*algorithm by testing alternative set-ups? Do you have any recommendations for the users, regarding the selection of these*
*inputs?*
        The user may choose to give more weight to some of the rules in the input arguments. We tested different combinations to
conclude that it is better to have a "balanced" approach with the same number of parameter sets generated for each rule (5 sets
for each rule for the default version).

*Page 7, section 4.2 (Hydrological modeling): Your case study does not allow extracting safe conclusions about the perfor-*
*mance of your method and its comparison against NSGA II. The key reason is that the use of a single overall metric, i.e. KGE,*
*ensures almost perfect fitting to observations (KGE = 95).*
        We chose to change the objective to get more variability.

*Page 8, line 8: Please, better explain criteria (2) and (3) and the associated signatures. Have been these criteria used*
*elsewhere? If yes, please also provide the associated references.*
        We changed these criteria to Nash-Sutcliffe and Kling-Gupta Efficiencies which are much more usual.

*Page 9, lines 3-4: "MCO has been used with crossover probability set to 0.5 and mutation probability to 0.3". Have you*
*made any preliminary tests before selective these values? Which are the values applied to the input arguments of caRamel?*
        We have run a sensitivity analysis with mco to chose the values. We will add the values of the input arguments for caRamel.

*Page 9, Figure 5: I find your figure a little bit misleading. In the vertical axis, the spread of solutions is very small, and within*
*the anticipated range of uncertainty induced in any hydrological calibration exercise. For instance, the lower value of KGEamd*
*is 0.83, while the higher is 0.86. From my point-of-view, such differences do not make sense in the real world.*
        We changed these criteria to Nash-Sutcliffe and Kling-Gupta Efficiencies at the oulet and at two interior gauging station to
have more variability.

*Page 10, line 12: How did you selected the best compromise parameter set? What do you mean by term "observed set"?*

We added: "To illustrate the results on the simulated stream flow, a "best-compromise set" has been selected regarding to the distance to the point (1,1,1) in the objective space for each hydrological case studies"

430

*Page 11, section 5 (Conclusions): This section is poorly developed. It has to be written from scratch, to highlight the advantages and weaknesses of the methodology and also discuss ideas for future research.*
We re-writed it.

435 *Minor editorial comments*
*Page 2, line 10: In which of the aforementioned papers do you describe the algorithm? It is not clear here.*
These papers refer to research work using caRamel but the algorithm itself was not described.

*Page 2, line 17: Term "Hydrology" should not start with capital.*
440 We corrected it.

*Page 2, line 18: Please, change "underconstrained" to read "unconstrained".*
We corrected it.

445 *Page 3, line 3: Please, change to read "with respect to".*
We corrected it.

*Page 8, line 16: Please, change to read "parameter sets".*
We corrected it.

450

*Page 9, line 7: Please, change to read "Pareto fronts".*
We corrected it

*Page 9, lines 16-17: Please, change to read "The GS metric exhibits a larger variability, thus a larger envelope for both*
455 *optimizers".*
We corrected it.

**References**

[revised manuscript text omitted]

---

## Referee Report (RR1)

**Review of "Multi-objective calibration by combination of stochastic and gradient-like parameter generation rules: the caRamel algorithm", by C. Monteil, F. Zaoui, N. Le Moine, and F. Hendrickx**

**Reviewer: Andreas Efstratiadis (andreas@itia.ntua.gr)**

**General comments and critique**

This version is the revision of the technical note that has been submitted to HESSD, under the title "The caRamel R package for Automatic Calibration by Evolutionary Multi Objective Algorithm". The content and philosophy of the revised article is very different, since the authors have made a large effort to provide a research paper rather than a short technical note. I am also very pleased for the detailed response letter, and the fact that many of my recommendations have been addressed.

I found this work much more comprehensive. The new algorithm is quite well presented, although I would expect a more careful review of the current advances in the field of multiobjective optimization and its applications in hydrological modelling, in order to justify the importance of this new methodology. In this context, I encourage the authors to address this important issue, which stands for any new research.

Another suggestion involves the presentation of results. I found them quite poor. To my opinion, the benchmark problem can be further developed, e.g. by running the three methods with much lower budget (instead of allowing 50 000 evaluations) and also changing the population size. You may take some ideas from the work by Tsoulakas et al. (2016), as well as many other similar works in the optimization literature. This will better reveal the pros and cons of the CaRamel method, and ensure a clearer comparison with respect to MEAS and NSGA-II.

Finally, I do think that a short section conclusive is missing, with guidance for optimal setting of algorithmic inputs.

There are also few additional comment and editorial correction, which are listed below.

In this respect, my overall recommendation is for a moderate revision.

**Specific comments**

Page 2, lines 32-33: "Most of multi-objective algorithms rely mainly on stochastic generation rules, with few deterministic aspects". This argument requires some development, since it denotes the motivation of your research. If possible, also add references.

Page 6, line 123: Please explain the meaning of "secondary optimum".

Page 6, section 3.1.3: Please, explain the criteria for selecting the so-called "a priori variance" of each parameter.

**Minor editorial comments**

Page 1, line 24: Please, change to read "have been" instead of "have become".

Page 2, line 25: Please, change to read "… problems that are too complex…"

Page 2, line 26: "The advantage of these evolutionary algorithms lies not only…." This sentence is unclear.

Page 2, line 31: Please, change to read "to meet the need for an automatic calibration".

Page 6, line 126: "… to make the variance of parameters independent from each other". This statement is unclear. Variance and independence are two different notions.

Page 6, line 139: It may be preferable using $M^T$ for transpose matrix. Most readers are familiar with this symbol.

Page 9, line 195: Please, open parenthesis.

**References**

Tsoukalas, I., P. Kossieris, A. Efstratiadis, and C. Makropoulos, Surrogate-enhanced evolutionary annealing simplex algorithm for effective and efficient optimization of water resources problems on a budget, Environmental Modelling and Software, 77, 122–142, doi:10.1016/j.envsoft.2015.12.008, 2016.

---

## Author Response (AR2)

We would like to thank the editor Elena Toth and the two referees Guillaume Thirel and Andreas Efstratiadis for their constructive suggestions and comments. Below we provide the Editor's and Referee's comments verbatim in black italic text and our responses below each comment in blue text.

**1    Response to editor decision**

*Dear Authors,*
*as I wrote already, this revised work is indeed a significant step forward, and it is almost ready for publication.*
Thank you for your careful reading.

*Nonetheless the referees have some constructive suggestions for improving it even more: in addition to make some technical minor changes suggested by both referees, Dr. Andreas Efstratiadis asks to provide a wider review of the state-of-the-art and to add a short conclusive section, and I am sure you may add such parts.*
Thank you for this suggestions, we added these two parts as you can see in our response to Dr Andreas Efstratiadis.

*Andreas also believes that adding an analysis on a different benchmark would help the readers to better assess the value of your algorithm: I would invite to consider if it is possible for you to provide such different benchmark, unless it takes too much time.*
In the current working conditions due to the covid crisis, it is not possible for us to run new computations. This benchmark is indeed a very interesting suggestion for further work.

*I do invite you, therefore, to submit a last revision trying to address such comments, in order to make your work even more appreciated by the HESS readers.*
Thank you, here is the response to the reviewers suggestions for this revised version.

**2    Response to suggestions for revision from Referee Andreas Efstratiadis**

*General comments and critique*
*This version is the revision of the technical note that has been submitted to HESSD, under the title "The caRamel R package for Automatic Calibration by Evolutionary Multi Objective Algorithm". The content and philosophy of the revised article is very different, since the authors have made a large effort to provide a research paper rather than a short technical note. I am also very pleased for the detailed response letter, and the fact that many of my recommendations have been addressed.*
Thank you for all these constructive recommendations.

*I found this work much more comprehensive. The new algorithm is quite well presented, although I would expect a more careful review of the current advances in the field of multiobjective optimization and its applications in hydrological modelling, in order to justify the importance of this new methodology. In this context, I encourage the authors to address this important issue, which stands for any new research.*
We added this paragraph: "In hydrology, Madsen (2003) have implemented automatic multi-objective calibration of MIKE SHE model (Refsgaard et Storm , 1995) on the Danish Karup catchment (440 $km^2$) with the SCE algorithm (Duan et al. , 1992). Yang et al. (2014) run a multi-objective optimization of the distributed hydrologic model MOBIDIC (Campo et al., 2006) on the Davidson catchment (North Carolina, 105 $km^2$) with the Non-dominated Sorting Genetic Algorithm II (NSGA-II, Deb et al., 2002). More recently Smith et al. (2019) lead a multi-objective ensemble approach to hydrological modelling in the UK over 303 catchments for historic drought reconstruction with GR4J conceptual model (Coron et al., 2017) by using Latin hypercube sampling (McKay et al. , 1979) and Pareto-optimising ranking approach accounting for non-acceptable trade-offs

45  (Efstratiadis and Koutsoyiannis, 2010). Mostafaie et al. (2018) have compared five different calibration techniques on GR4J lumped hydrological model using in situ runoff and daily data from the Gravity Recovery And Climate Experiment (GRACE, Tapley et al. , 2004). They conclude that according to the diversity based metrics NSGA-II method is the best one, according to the accuracy metric Multi-objective Particle Swarm Optimization (MPSO,Reddy and Nagesh Kumar , 2007) is ranked first and finally, the performance of all algorithms is found the same, while considering the cardinality measure."

50

*Another suggestion involves the presentation of results. I found them quite poor. To my opinion, the benchmark problem can be further developed, e.g. by running the three methods with much lower budget (instead of allowing 50 000 evaluations) and also changing the population size. You may take some ideas from the work by Tsoukalas et al. (2016), as well as many other similar works in the optimization literature. This will better reveal the pros and cons of the CaRamel method, and ensure a*
55  *clearer comparison with respect to MEAS and NSGA-II.*
We added more comments to improve our presentation of results. The benchmark strategy that you suggest is interesting. Unfortunatelywe can't conduct it, specifically in the current context where we don't have access to our computational resources. Thus we suggest to mention it as a perspective in our optimization evaluation framework.
"We chose to run an important number of model evaluations and optimizations to get representative results and assess the re-
60  producibility of the optimization. Others benchmark methodology would be conceivable, such as presented by Tsoukalas et al. (2016) where several test functions and two water resources applications are implemented to compare the Surrogate-Enhanced Evolutionary Annealing Simplex algorithm (SEEAS) to four other mono-objective optimization algorithms. In this study, two alternative computational budget (indicated by the maximal number of model evaluations) are considered which impacts the paramaters of the optimizers."

65

*Finally, I do think that a short section conclusive is missing, with guidance for optimal setting of algorithmic inputs.*
We added:" An optimization algorithm might be delicate to use because of the choice of input arguments which are specific to the algorithm and might require some "expert knowledge". The sensitivity to caRamel internal parameters has not been presented in this manuscript, but we have done some sensitivity analysis with the Morris method (Morris, 1991) to recommend
70  some default values for the user. First it is recommended to give the same weight to each generation rule by indicating the same number of parameter sets to generate. It is interesting to generate a small number of sets by generation to reduce the number of model evaluations and have a more rapid convergence. By default, five sets are generated for each rule. The size of the initial population should be large enough to have enough variability (at least 50 sets for a complex model). Moreover, as convergence can be sensitive to the randomly chosen initial population, it is recommended to run two or three optimizations to
75  assess reproducibility. "

*There are also few additional comment and editorial correction, which are listed below. In this respect, my overall recommendation is for a moderate revision.*

80  *Specific comments*
*Page 2, lines 32-33: "Most of multi-objective algorithms rely mainly on stochastic generation rules, with few deterministic aspects". This argument requires some development, since it denotes the motivation of your research. If possible, also add references.*
We corrected: "Madsen (2003) indicates that the global population-evolution-based algorithms are more effective than multi-
85  start local search procedures, which in turn perform better than pure local search methods. However most of multi-objective algorithms rely mainly on stochastic generation rules, with few deterministic aspects, as it is the case in the widely used NSGA-II for instance."

*Page 6, line 123: Please explain the meaning of "secondary optimum".*
90  We corrected as "local optimum".

*Page 6, section 3.1.3: Please, explain the criteria for selecting the so-called "a priori variance" of each parameter.*
We added: "The a priori variance is computed for each parameter from the bounds of variation indicated as input of the optimizer."

*Minor editorial comments*
*Page 1, line 24: Please, change to read "have been" instead of "have become".* ok

*Page 2, line 25: Please, change to read "… problems that are too complex…"* ok

*Page 2, line 26: "The advantage of these evolutionary algorithms lies not only…." This sentence is unclear.*
We rephrased it as: "The advantage of these evolutionary algorithms lies not only because there are few alternatives for searching substantially large spaces for multiple Pareto-optimal solutions but also due their inherent parallelism and capability to exploit similarities of solutions by recombination, that enables them to approximate the Pareto-optimal front in a single optimization run."

*Page 2, line 31: Please, change to read "to meet the need for an automatic calibration".* ok

*Page 6, line 126: "… to make the variance of parameters independent from each other". This statement is unclear. Variance and independence are two different notions.*
We reprased it as: "to make the parameters varying independently from each other."

*Page 6, line 139: It may be preferable using $M^T$ for transpose matrix. Most readers are familiar with this symbol.* ok, we used it.

*Page 9, line 195: Please, open parenthesis.* ok

**3 Response to suggestions for revision from Referee Guillaume Thirel**

*This is my second review of this manuscript.*
*I read with attention the new version of the manuscript and the answers to the reviewers and editor comments. The authors made a very serious job in taking into account all the comments.*
*First, the format of the manuscript complies now much more with the format of the HESS article. This is not anymore the description of a package, but the description and analysis of a calibration method, using an R package. As such, I feel it belongs now clearly to the research article category of HESS. The title was modified accordingly. The description of the algorithm was extended. Not sure whether it is too long, in any case I do not have suggestions for shortening. A new algorithm was used for comparison and this comparison was extended. As a consequence, I would be in favor of publication.*
Thank you for the helpful suggestions you made to us and for these nice comments.

*I only have a couple of minor remarks:*
*Abstract: some suggestions: "add" -> "adding"; "find" -> "finding"; "low-constrained" -> "poorly-constrained"* ok

*Regarding the goals of the paper (lines 44-47): I would add the information that pieces of codes are provided in the appendix.*
We added this sentence in the introduction: "Pieces of codes are provided in the appendix."

*I would also replace "for each case study" with something like "for an analytical example and for three river case studies", as these case studies were not introduced yet at this point of the manuscript.* ok, we replaced it.

*L. 101: proper definitions of "vertex" and "simplex" could be useful for newcomers to multi-objective optimization.*
We added: "For the rules 1 and 2, we use the notion of simplex which is a generalization of the notion of a triangle to higher dimensions: a 0-simplex is a point,a 1-simplex a line segment, a 2-simplex is triangle, a 3-simplex a tetrahedron. A vertex is a point where two or more edges meet."

 *L. 239 (and following): the Blue River is a real catchment and data are real.*

We were referring to the documentation of `airGR` package: "L0123001, L0123002 or L0123003 are fictional catchments." We corrected it in the paper.

 *L. 274-275: I would move the sentence "The theoretical..." after "at Montdauphin)" in line 279 for better readability.* ok, we moved it.

*L. 289: "changeS"* ok

 *In the document we have both "discharges" and "stream flow" terms. I would pick one and stick to it.*
We corrected all "stream flow" terms by "discharges".

*Figure 10: the GR4J parameter values are provided in the transformed space. I suggest transforming them back to the real space through the TransfoParam_GR4J function and I would also suggest adding the units of these parameters. As the trans-*
 *formations are not linear, the boxplots should somehow be modified.*
Figure 10 displays now the parameters whith theirs units.

*Caption figure 10: "Durance River"* ok

 *L. 353: "parameter sets"* ok

*Appendix C: several R assignations ("<-") were wrongly modified to "«-".* "«-" is used to declare global variables.

*Also, line 411, the square sign is erroneous.* ok, we corrected it.

*References: Mersmann et al.: please add the package version you used.* ok, it is version 1.0-15.1

**References**

[revised manuscript text omitted]

---

## Author Response (AR3)

**1 Response to editor decision**

We would like to thank the editor Elena Toth and the two referees Guillaume Thirel and Andreas Efstratiadis for their constructive suggestions and comments during all the revision phase. Below we provide the Editor's comments verbatim in black italic text and our responses below each comment in blue text.

*Dear Authors,*
*many thanks for having clarified/integrated the last few points as suggested by the Referees.*
Thank you for your careful reading.

*We fully understand that, especially due to this difficult period, it is not possible for you to perform additional computations, that may be carried out ion a future study.*
*The phrase at l. 26 p. 2 is still not clear, due to the English phrasing: i would suggest: "Evolutionary algorithms are advantageous not only because...."?*
We rephrased it as : "Evolutionary algorithms are advantageous not only because there are few alternatives for searching substantially large spaces for multiple Pareto-optimal solutions, but also due to their inherent parallelism and capability to exploit similarities of solutions by recombination that enables them to approximate the Pareto-optimal front in a single optimization run".

*Many thanks for having shared your algorithm with the HESS community: I hope it will be widely used.*
Thank you.

**2 List of relevant changes**

We corrected the above-mentioned sentence and figures and tables have been placed at the end of the manuscript.

**3 marked-up manuscript version**

[revised manuscript text omitted]